# DLGNet: Hyperedge Classification through Directed Line Graphs for Chemical Reactions

## Abstract

Graphs and hypergraphs provide powerful abstractions for modeling interactions among a set of entities of interest and have been attracting a growing interest in the literature thanks to many successful applications in several fields. In particular, they are rapidly expanding in domains such as chemistry and biology, especially in the areas of drug discovery and molecule generation. One of the areas witnessing the fasted growth is the chemical reactions field, where chemical reactions can be naturally encoded as directed hyperedges of a hypergraph. In this paper, we address the chemical reaction classification problem by introducing the notion of a *Directed Line Graph* (DLG) associated with a given directed hypergraph. On top of it, we build the Directed Line Graph Network (DLGNet), the first spectral-based Graph Neural Network (GNN) expressly designed to operate on a hypergraph via its DLG transformation. The foundation of DLGNet is a novel Hermitian matrix, the *Directed Line Graph Laplacian* $\vec{\mathbb{L}}_N$, which compactly encodes the directionality of the interactions taking place within the directed hyperedges of the hypergraph thanks to the DLG representation. $\vec{\mathbb{L}}_N$ enjoys many desirable properties, including admitting an eigenvalue decomposition and being positive semidefinite, which make it well-suited for being adopted within a spectral-based GNN. Through extensive experiments on chemical reaction datasets, we show that DLGNet significantly outperforms the existing approaches, achieving on a collection of real-world datasets an average relative-percentage-difference improvement of 3.27%, with a maximum improvement of 5.28%.

## 1 Introduction

In recent years, ground-breaking research in the graph-learning literature has been prompted by seminal works on GNNs such as Scarselli et al. (2009); Micheli (2009); Li et al. (2016); Kipf &Welling (2017); Veličković et al. (2018). However, representing data solely through graphs, either undirected or directed, can be limiting in many real-world applications where more complex relationships exist. In such cases, generalizations of graphs known as hypergraphs, which allow for higher-order (group) relationships among the vertices, have emerged as powerful alternatives. Hypergraphs extend the traditional concept of a graph by allowing *hyper*edges to connect an arbitrary number of nodes, thereby capturing both pairwise (dyadic) and group-wise (polyadic) interactions (Schaub et al., 2021). This has naturally led to a new stream of research devoted to the investigation of Hypergraph Neural Networks (HNNs) (Feng et al., 2019; Chien et al., 2021; Huang &Yang, 2021; Wang et al., 2023a;b).

Among many successful applications, graph and hypergraph representations have recently been applied in chemistry and biology to address various tasks such as drug discovery (Bongini et al., 2021), molecule generation (Hoogeboom et al., 2022), and protein interaction modeling (Jha et al., 2022). Several graph-based representations have also been developed and employed for the study of chemical reactions, which has applications in areas such as reaction engineering, retrosynthetic pathway design, and reaction feasibility evaluations. In particular, retrosynthetic modeling, where a synthetic route is designed starting from the desired product and analyzed backward, benefits greatly from accurate reaction type identification. This capability enables the elimination of unfeasible pathways, thereby streamlining the discovery of efficient routes for chemical production. This is particularly important in industries such as pharmaceutical and material sciences, where optimizing synthetic routes can lead to significant cost savings and innovation. A similar situation holds, in

reaction feasibility analysis, where predicting the likelihood of a reaction's success based on the molecular inputs is essential for designing scalable and efficient processes.

One of the most relevant techniques to model reactions is the directed graph (Fialkowski et al., 2005), where molecules are represented as nodes and the chemical reactions are represented as directed edges from reactants to products. Despite its popularity, such a directed graph model suffers from some key limitations. In particular, modeling each reaction as a collection of *individual* directed edges between each reactant-product pair fails to fully capture the complexity of multi-reactant or multi-product reactions, which are key to many important applications (Restrepo, 2022; Garcia-Chung et al., 2023). To mitigate this issue, Chang (2024) proposed a hypergraph representation in which molecules are nodes and each reaction is captured by a hyperedge. However, this model lacks a mechanism to represent the directionality of a reaction, thus failing to capture the reactant/product relationship within it. As a further attempt, Restrepo (2023) introduced a directed hypergraph representation which is able to model both the chemical reactions structure and their directionality, where directed hyperedges model the directional interactions between reagents (heads) and products (tails), better capturing the full complexity of chemical reactions. Let us note that this literature only focuses on modeling reaction structures without considering any form of hypergraph learning methods. We set ourselves out to developing one in this paper.

In contrast to prior studies that address node classification or link prediction tasks (Dong et al., 2020; Wang et al., 2023b; Zhao et al., 2024), in this work we tackle the reaction classification problem (i.e., the problem of predicting the reaction type of a given set of reactants and products) as a *hyperedge classification* task. This task can be reframed as a node classification task by using the concept of the line graph.

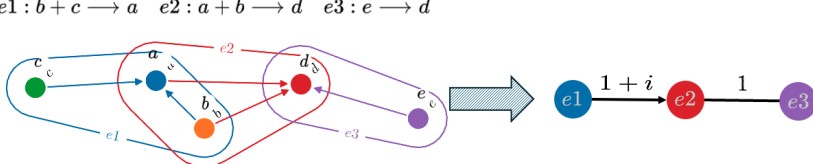

Figure 1: Three reactions and their corresponding directed hypergraph representation (left). The hypergraph is then transformed into its directed line graph (right). The hyperedges of $\vec{H}$ become the nodes of $\mathrm{DLG}(\vec{H})$m and are connected if they overlap in $\vec{H}$. Complex-valued edge weights in $\mathrm{DLG}(\vec{H})$ encode $\vec{H}$'s directionality, as detailed in Section 3.

With this goal in mind, we introduce the concept of a directed line graph of a given directed hypergraph $\vec{H}$: the *Directed Line Graph* $\mathrm{DLG}(\vec{H})$. In $\mathrm{DLG}(\vec{H})$, the vertices correspond to the hyperedges of $\vec{H}$, and a directed edge connects two vertices if the corresponding hyperedges in $\vec{H}$ share at least a vertex, as shown in Figure 1. Since the nodes of $\mathrm{DLG}(\vec{H})$ correspond to hyperedges of $\vec{H}$, this modeling approach allows us to directly operate on hyperedge features, which are critical for solving the reaction-classification task. To this end, we define the *Directed Line-Graph Laplacian* $(\vec{\mathbb{L}}_N)$, a Laplacian matrix which is specifically designed to capture both directed and undirected adjacency relationships between the hyperedges of $\vec{H}$ via its directed line graph $\mathrm{DLG}(\vec{H})$. We prove that $\vec{\mathbb{L}}_N$ enjoys different key properties, among which being Hermitian (i.e., being a complex-valued matrix with a symmetric real part and a skew-symmetric imaginary one) and positive semidefinite. These properties allows us to introduce a spectral convolutional operator for $\mathrm{DLG}(\vec{H})$. Thanks to the association of $\mathrm{DLG}(\vec{H})$ with the original directed hypergraph $\vec{H}$, $\mathrm{DLG}(\vec{H})$ serves as the foundation of Directed Line Graph Network (DLGNet), the first (to our knowledge) spectral-based GNN designed for the convolution of hyperedge features rather than node features.

For the task of hyperedge classification for the prediction of chemical reaction classes, transitioning from a directed hypergraphs to a directed line graph representation proves to offer significant advantages, as evidenced by our experimental results. Specifically, DLGNet achieves an average relative percentage difference improvement of 3.27% over the second-best method across a collection of real-world datasets, with a maximum improvement of 5.28%. We also carry out an extensive set of ablation studies, which confirm the importance of the various components of DLGNet.

**Main Contributions of This Work**

- We introduce the first formal definition of a directed line graph associated with a directed hypergraph $\vec{H}$: the *Directed Line Graph* $\text{DLG}(\vec{H})$.

- We propose the Directed Line Graph Laplacian $\vec{\mathbb{L}}_N$, a Hermitian matrix that captures both directed and undirected relationships between the hyperedges of a directed hypergraph via its DLG. We also prove that $\vec{\mathbb{L}}_\mathbb{N}$ possesses many desirable spectral properties.

- We introduce DLGNet, the first spectral-based Graph Neural Network specifically designed to operate on directed line graphs associated with directed hypergraphs by directly convolving hyperedge features rather than node features.

- We perform an extensive experimental evaluation on the chemical reaction-classification task on real-world datasets. Our results highlight the advantages of our approach compared to other methods presented in the literature.

## 2 BACKGROUND

An undirected hypergraph is defined as an ordered pair $H = (V, E)$, with $n := |V|$ and $m := |E|$, where $V$ is the set of vertices (or nodes) and $E \subseteq 2^V \setminus \{\}$ is the (nonempty) set of hyperedges. The weights of the hyperedges are stored in the diagonal matrix $W \in \mathbb{R}^{m \times m}$, where $w_e$ is the weight of hyperedge $e \in E$ (in the unweighted case we have $W = I$). The vertex degree $d_u$ and hyperedge degree $\delta_e$ are defined as $d_u := \sum_{e \in E : u \in e} |w_e|$ for $u \in V$, and $\delta_e := |e|$ for $e \in E$. These degrees are stored in two diagonal matrices $D_v \in \mathbb{R}^{n \times n}$ and $D_e \in \mathbb{R}^{m \times m}$. In the case of 2-uniform hypergraphs, the matrix $A \in \mathbb{R}^{n \times n}$ is defined such that $A_{uv} = w_e$ for each $e = \{u, v\} \in E$ and $A_{uv} = 0$ otherwise; we refer to it as the *adjacency* matrix of the graph. Hypergraphs where $\delta(e) = k$ for some $k \in \mathbb{N}$ for all $e \in E$ are called $k$-uniform. Graphs are 2-uniform hypergraphs. Following Gallo et al. (1993), we define a directed hypergraph $\vec{H}$ as a hypergraph where each node in each hyperedge $e \in E$ belongs to either a *head set* $H(e)$ or a *tail set* $T(e)$. If $T(e)$ is empty, $e$ is an undirected hyperedge.

The relationship between vertices and hyperedges in a undirected hypergraph $H$ is classically represented via an incidence matrix $B$ of size $|V| \times |E|$, where $B$ is defined as:

$$B_{ve} = \begin{cases} 1 & \text{if } v \in e \\ 0 & \text{otherwise} \end{cases} \qquad v \in V, e \in E. \tag{1}$$

From the incidence matrix $B$, one can derive the *Signless Laplacian Matrix* $Q$ as well as its normalized version $Q_N$ (Chung &Graham, 1997):

$$Q := BWB^\top \qquad\qquad Q_N := D_v^{-\frac{1}{2}} BW D_e^{-1} B^\top D_v^{-\frac{1}{2}}, \tag{2}$$

where $W, D_e, D_v$ are the diagonal matrices defined above. Following Zhou et al. (2006), the Laplacian for a general undirected hypergraph is defined as follows:

$$\Delta := I - Q_N. \tag{3}$$

The Laplacian matrix encodes the hypergraph's connectivity and hyperedge weights.

### 2.1 GRAPH FOURIER AND GRAPH CONVOLUTIONS

Let $\mathcal{L}$ be a generic Laplacian matrix of a given 2-uniform hypergraph $H$. We assume that $\mathcal{L}$ admits an eigenvalue decomposition $\mathcal{L} = U\Lambda U^*$, where $U \in \mathbb{C}^{n \times n}$ represents (in its columns) the eigenvectors, $U^*$ is its conjugate transpose, and $\Lambda \in \mathbb{R}^{n \times n}$ is the diagonal matrix containing the eigenvalues. Let $x \in \mathbb{C}^n$ be a *graph signal*, i.e., a complex-valued function $x : V \to \mathbb{C}^n$ of the vertices of $H$. We define $\hat{x} = \mathcal{F}(x) = U^*x$ as the *graph Fourier transform* of $x$ and $\mathcal{F}^{-1}(\hat{x}) = U\hat{x}$ its inverse transform. The convolution $y \circledast x$ between $x$ and another graph signal $y \in \mathbb{C}^n$, acting as a *filter*, in the vertex space is defined in the frequency space as $y \circledast x = U\text{diag}(U^*y)U^*x$. Letting $\hat{Y} := U\hat{G}U^*$ with $\hat{G} := \text{diag}(U^*y)$, we can write $y \circledast x$ as the linear operator $\hat{Y}x$. See Shuman et al. (2013) for more details.

In the context of GNNs, explicitly learning $y$ as a *non-parametric filter* presents two significant limitations. Firstly, computing the eigenvalue decomposition of $\mathcal{L}$ can be computationally too expensive (Kipf &Welling, 2017). Secondly, explicitly learning $y$ requires a number of parameters proportional to the input size, which becomes inefficient for high-dimensional tasks (Defferrard et al., 2016). To address these issues, the GNN literature commonly employs filters where the graph Fourier transform is approximated as a degree-$K$ polynomial of $\Lambda$, with $K$ kept small for computational efficiency. For further details, we refer the reader to Kipf &Welling (2017); Defferrard et al. (2016); Huang et al. (2024). This leads to a so-called *localized filter*, thanks to which the output (i.e., filtered) signal at a vertex $u \in V$ is a linear combination of the input signals within $K$ edges of $u$ (Shuman et al., 2013). By employing various polynomial filters and setting $K = 1$ (as commonly employed in the literature), such as Chebyshev polynomials as in Hammond et al. (2011); Kipf &Welling (2017) or power monomials as used by Singh &Chen (2022), one obtains a parametric family of linear operators with two learnable parameters, $\theta_0$ and $\theta_1$:[1]

$$\hat{Y} := \theta_0 I + \theta_1 \mathcal{L}. \tag{4}$$

## 3 THE DIRECTED LINE GRAPH AND ITS LAPLACIAN

The *line graph* $L(H)$ of a generic undirected hypergraph $H$ is classically defined as the undirected graph whose vertex set is the hyperedge set of $H$. In $L(H)$, two vertices $i, j$ are adjacent—i.e., $L(H)$ contains the edge $\{i, j\}$—if and only if their corresponding hyperedges $i, j$ have a nonempty intersection (Tyshkevich &Zverovich, 1998). By construction, $L(H)$ is a 2-uniform graph. Its adjacency matrix is defined as:

$$A(L(H)) := \mathbb{Q} - W D_e, \tag{5}$$

where $\mathbb{Q} := B^\top B$ is, by construction, the Signless Laplacian of $L(H)$.[2] The normalized version of $\mathbb{Q}$ and the corresponding normalized Laplacian are defined as:

$$\mathbb{Q} := W^{\frac{1}{2}} B^\top B W^{\frac{1}{2}} \qquad \mathbb{Q}_N := D_e^{-\frac{1}{2}} W^{\frac{1}{2}} B^\top D_v^{-1} B W^{\frac{1}{2}} D_e^{-\frac{1}{2}} \qquad \mathbb{L}_N := I - \mathbb{Q}_N. \tag{6}$$

Notice that, from equation 2, one can define the weighted version of $B$ as $B W^{\frac{1}{2}}$. The definitions in equation 6 rely on the same matrix, but transposed.

To the best of our knowledge, the literature does not offer any formal definition for the line graph associated with a (weighted) directed hypergraph $\vec{H}$ (it does only for the undirected case). The availability of such a definition could be crucial for tasks where the hyperedge direction is important.

To address this gap, we first define a complex-valued incidence matrix $\vec{B}$ which preserves the inherent directionality of $\vec{H}$:

$$\vec{B}_{ve} := \begin{cases} 1 & \text{if } v \in H(e), \\ -\mathrm{i} & \text{if } v \in T(e), \qquad v \in V, e \in E. \\ 0 & \text{otherwise.} \end{cases} \tag{7}$$

Building on $\vec{B}$, we propose the following definition for the directed line graph associated with a directed hypergraph $\vec{H}$:

**Definition 1.** *The Directed Line Graph $DLG(\vec{H})$ of a directed hypergraph $\vec{H}$ is a 2-uniform hypergraph whose vertex set corresponds to the hyperedge set of $\vec{H}$ and whose adjacency matrix is the following complex-valued skew-symmetric matrix:*

$$A(DLG(\vec{H})) = W^{\frac{1}{2}} \vec{B}^* \vec{B} W^{\frac{1}{2}} - W D_e. \tag{8}$$

Using equation 8 of definition 1 and equations 5–6, we obtain the following formulas for the normalized Signless Laplacian $\vec{\mathbb{Q}}_N \in \mathbb{C}^{m \times m}$ and the normalized Laplacian $\vec{\mathbb{L}}_N \in \mathbb{C}^{m \times m}$ of $DLG$, which we refer to by *Signless Directed Line-Graph* and *Directed Line Graph Laplacian*:

$$\vec{\mathbb{Q}}_N := \vec{D}_e^{-\frac{1}{2}} W^{\frac{1}{2}} \vec{B}^* \vec{D}_v^{-1} \vec{B} W^{\frac{1}{2}} \vec{D}_e^{-\frac{1}{2}} \qquad \vec{\mathbb{L}}_N := I - \vec{\mathbb{Q}}_N. \tag{9}$$

---

[1]Following w.l.o.g. Singh &Chen (2022), we employ the approximation $\hat{G} = \sum_{k=0}^{K} \theta_k \Lambda^k$, from which we deduce $\hat{Y}x = U\hat{G}U^*x = U(\sum_{k=0}^{K} \theta_k \Lambda^k)U^*x = \sum_{k=0}^{K} \theta_k (U\Lambda^k U^*)x = \sum_{k=0}^{K} \theta_k \mathcal{L}^k x$.

[2]This follows from the fact that the incidence matrix of $L(H)$ is $B^*$.

To better understand how $\vec{\mathbb{L}}_N$ encodes the directionality of $\vec{H}$, we illustrate its definition in scalar form for a pair of hyperedges $i, j \in E$ (which correspond to vertices in $DLG(\vec{H})$):

$$
\vec{\mathbb{L}}_N(ij) = \begin{cases} 1 - \sum_{u \in i} \dfrac{w_i}{d_u \delta_i} & i = j \\[2em] \left( - \sum_{\substack{u \in H(i) \cap H(j) \\ \vee u \in T(i) \cap T(j)}} \dfrac{w_i^{\frac{1}{2}} w_j^{\frac{1}{2}}}{d_u} - \mathrm{i} \left( \sum_{u \in H(i) \cap T(j)} \dfrac{w_i^{\frac{1}{2}} w_j^{\frac{1}{2}}}{d_u} - \sum_{u \in T(i) \cap H(j)} \dfrac{w_i^{\frac{1}{2}} w_j^{\frac{1}{2}}}{d_u} \right) \right) \dfrac{1}{\delta_i^{\frac{1}{2}}} \dfrac{1}{\delta_j^{\frac{1}{2}}} & i \neq j \end{cases} \tag{10}
$$

When $i = j$, we are in the self-loop part of the equation and $\vec{\mathbb{L}}_N(ij)$ weights hyperedge $i$ proportionally to its weight $w_i$ and inversely proportionally to its density and the density of its nodes. When $i \neq j$, $\vec{\mathbb{L}}_N(ij)$'s value depends on the interactions between the hyperedges of $\vec{H}$ (which correspond to the nodes of $DGL(\vec{H})$). Let $u \in V$ be a node and $i, j \in E$ be two hyperedges in the hypergraph $\vec{H}$. If $u$ belongs to the head set of both the hyperedges (i.e., $u \in H(i) \cap H(j)$) or to the tail set of both (i.e., $u \in T(i) \cap T(j)$), its contribution to the real part of $\mathbb{L}_N(ij)$, $\Re(\vec{\mathbb{L}}_N(ij))$, is negative. For the undirected line graph associated with an undirected hypergraph, this is the only contribution, consistent with the behavior of $\mathbb{L}_N$ (as described in equation 6). If $u$ takes opposite roles in hyperedges $i$ and $j$, i.e, it belongs to the head set in $i$ and to the tail set in $j$ or *vice versa*, it contributes to the imaginary part of $\mathbb{L}_N$, $\Im(\vec{\mathbb{L}}_N(ij))$, negatively when $u \in H(i) \cap T(j)$, and positively when $u \in T(i) \cap H(j)$. Consequently $\Im(\vec{\mathbb{L}}_N(ij))$ coincides with the *net* contribution of all the vertices that are shared between the hyperedges $i$ and $j$. An example illustrating the construction of $\vec{\mathbb{L}}_N$ for a directed line graph associated with a directed hypergraph is provided in Appendix G. Let us point out that the behavior of *Directed Line Graph Laplacian* differs from every (to the best of our knowledge) Laplacian matrix previously proposed in literature (see Appendix B for more details).

With the following theorem, we show that $\vec{\mathbb{L}}_N$ is a generalization of $\mathbb{L}_N$ (defined in equation 6) from the undirected to the directed case:

**Theorem 1.** *If $\vec{H}$ is undirected (i.e., $\vec{H} = H$), $\vec{\mathbb{L}}_N = \mathbb{L}_N$ and $\vec{\mathbb{Q}}_N = \mathbb{Q}_N$ holds.*

The *Directed Line Graph Laplacian* enjoys several properties. First, to be able to adopt our Laplacian within a convolution operator in line with Kipf &Welling (2017) and other literature approaches (Zhang et al., 2021; Fiorini et al., 2023), we must show that our Laplacian is positive semidefinite. For this, we work out the expression for the 2-Dirichlet energy function associated with it. Such a function coincides with the Euclidean norm $||x||^2_{\vec{\mathbb{L}}_N}$ induced by $\vec{\mathbb{L}}_N$ for a signal $x \in \mathbb{C}^m$:

**Theorem 2.** *Letting $\mathbf{1}$ be the indicator function, the Euclidean norm induced by $\vec{\mathbb{L}}_N$ of a complex-valued signal $x = a + ib \in \mathbb{C}^m$ with a component per hyperedge in $E$ reads:*

$$
\frac{1}{2} \sum_{u \in V} \frac{1}{d(u)} \sum_{i,j \in E} \left( \left( \frac{w(j)^{\frac{1}{2}} a_i}{\delta(i)^{\frac{1}{2}}} - \frac{w(i)^{\frac{1}{2}} a_j}{\delta(j)^{\frac{1}{2}}} \right)^2 + \left( \frac{w(j)^{\frac{1}{2}} b_i}{\delta(i)^{\frac{1}{2}}} - \frac{w(i)^{\frac{1}{2}} b_j}{\delta(j)^{\frac{1}{2}}} \right)^2 \right) \mathbf{1}_{u \in H(i) \cap H(j) \vee u \in T(i) \cap T(j)}
$$

$$
+ \left( \left( \frac{w(j)^{\frac{1}{2}} a_i}{\delta(i)^{\frac{1}{2}}} - \frac{w(i)^{\frac{1}{2}} b_j}{\delta(j)^{\frac{1}{2}}} \right)^2 + \left( \frac{w(i)^{\frac{1}{2}} a_j}{\delta(j)^{\frac{1}{2}}} + \frac{w(j)^{\frac{1}{2}} b_i}{\delta(i)^{\frac{1}{2}}} \right)^2 \right) \mathbf{1}_{u \in H(i) \cap T(j)}
$$

$$
+ \left( \left( \frac{w(j)^{\frac{1}{2}} a_i}{\delta(i)^{\frac{1}{2}}} + \frac{w(i)^{\frac{1}{2}} b_j}{\delta(j)^{\frac{1}{2}}} \right)^2 + \left( \frac{w(i)^{\frac{1}{2}} a_j}{\delta(j)^{\frac{1}{2}}} - \frac{w(j)^{\frac{1}{2}} b_i}{\delta(i)^{\frac{1}{2}}} \right)^2 \right) \mathbf{1}_{u \in T(i) \cap H(j)}. \tag{11}
$$

Since the function in Theorem 2 is a real-valued sum of squares, we deduce the following spectral property for $\vec{\mathbb{L}}_N$:

**Corollary 1.** $\vec{\mathbb{L}}_N$ *is positive semidefinite.*

From equation 9, we have that $\vec{\mathbb{L}}_N = I - \vec{\mathbb{Q}}_N$. Thanks to Theorem 3, we show next that $\vec{\mathbb{Q}}_N$ has a nonnegative spectrum:

**Theorem 3.** $\vec{\mathbb{Q}}_N$ *is positive semidefinite.*

By applying Theorem 3 and Corollary 1, we can derive upper bounds on the spectra of $\vec{\mathbb{L}}_N$ and $\vec{\mathbb{Q}}_N$:

**Corollary 2.** $\lambda_{\max}(\vec{\mathbb{L}}_N) \leq 1$ and $\lambda_{\max}(\vec{\mathbb{Q}}_N) \leq 1$.

The proofs of the theorems and corollaries of this section can be found in Appendix B.

## 4 THE DIRECTED LINE GRAPH NETWORK (DLGNET)

The properties of the proposed Laplacian make it possible to derive a well-defined spectral convolution operator from it. In this work, this operator is integrated into the Directed Line Graph Network (DLGNet). Specifically, based on equation 4, by setting $\mathcal{L} = \vec{\mathbb{L}}_N$, the convolution operator is defined as $\hat{Y}x = \theta_0 I + \theta_1 \vec{\mathbb{L}}_N$. The advantage of adopting two parameters $\theta_0, \theta_1$ within DLGNet's localized filter is explained by the following result:

**Proposition 1.** *The convolution operator derived from equation 4 by setting $\mathcal{L} = \vec{\mathbb{L}}_N$ with parameters $\theta_0$ and $\theta_1$ is the same as the convolution operator obtained by using $\mathcal{L} = \vec{\mathbb{Q}}_N$ with parameters are rewritten as $\theta'_0 = \theta_0 + \theta_1$ and $\theta'_1 = -\theta_1$.*

This shows that DLGNet, by selecting appropriate values for $\theta_0$ and $\theta_1$, can leverage either $\vec{\mathbb{L}}_N$ or $\vec{\mathbb{Q}}_N$ as convolution operator to maximize the performance on the task at hand.

We define $X \in \mathbb{C}^{m \times c_0}$ as a $c_0$-dimensional graph signal (a graph signal with $c_0$ input channels), which we compactly represent as a matrix. This matrix serves as the feature matrix of the hyperedges of $\vec{H}$ which we construct from the feature matrix of the nodes $X' \in \mathbb{C}^{n \times c_0}$ of $\vec{H}$. Specifically, inspired by the operation used in the *reduction component* for graph pooling (Grattarola et al., 2022), we define the feature matrix for the vertices of $DGL(\vec{H})$ as $X = \vec{B}^* X'$. This approach combines features through summation, based on the topology defined by $\vec{B}$. See Appendix E for more details.

In our network, the scalar parameters $\theta_0$ and $\theta_1$ are subsumed by two operators $\Theta_0, \Theta_1 \in \mathbb{C}^{c_0 \times c}$ which we use to carry out a linear transformation on the feature matrix $X$. A similar transformation, which can also increase or decrease the number of channels of $X$, is adopted in other GNNs such as MagNet (Zhang et al., 2021). DLGNet features $\ell$ convolutional layers. The output $Z \in \mathbb{C}^{m \times c'}$ of any such layer adheres to the following equation:

$$Z(X) = \phi\left(IX\Theta_0 + \vec{\mathbb{L}}_N X\Theta_1\right), \tag{12}$$

where $\phi$ is the activation function. Following (Fiorini et al., 2023; 2024), DLGNet employs a complex *ReLU* where $\phi(z) = z$ if $\Re(z) \geq 0$ and $\phi(z) = 0$ otherwise, with $z \in \mathbb{C}$. DLGNet also utilizes a residual connection for every convolutional layer except the first one, a choice which helps prevent oversmoothing and has been proven to be helpful in a number of works, including (He et al., 2016; Kipf &Welling, 2017). After the convolutional layers, following Zhang et al. (2021), we apply an *unwind* operation where we transform $Z(X) \in \mathbb{C}^{m \times c'}$ into $(\Re(Z(X)) || \Im(Z(X))) \in \mathbb{R}^{m \times 2c'}$, where $||$ is the concatenation operator. To obtain the final results, DLGNet features $S$ linear layers, with the last one employing a Softmax activation function. DLGNet's architecture is depicted in Figure 2.

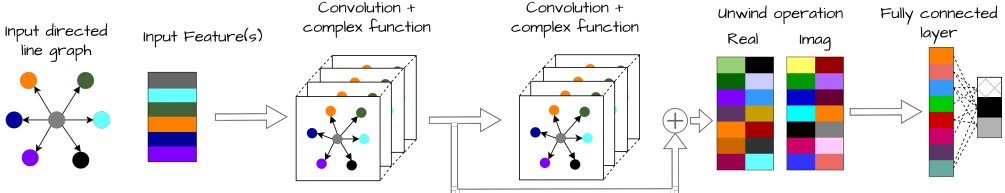

Figure 2: DLGNet's architecture: after two complex convolutional layers and a residual connection, we unwind the real and imaginary parts of the feature matrix and apply a fully connected layer.

**Complexity of DLGNet.** Let us assume (w.l.o.g.) that each of DLGNet's convolutional layers has $c$ input and output channels, while the last layer has $c$ input and $c'$ output channels ($c'$ is also the number of input channel to the linear layers). Let $d$ be number of output channel of the last linear layer (where

$d$ is the number of classes to be predicted). With $\ell$ convolutional layers and $S$ linear layers, DLGNet's complexity is $O(mnc_0) + O(\ell(m^2c + mc^2) + mc + (S-1)(mc'^2) + mc'd + md)$. Assuming $O(c) = O(c') = O(d) = \bar{c}$, we have a complexity of $O(\ell(m^2\bar{c}) + (\ell + S)(m\bar{c}^2))$. This shows that DLGNet has a quadratic complexity w.r.t. the number of hyperpedges $m$ and the asymptotic number of channels $\bar{c}$. For more details, see Appendix C.

## 5 EXPERIMENTAL RESULTS

We present three real-world datasets, the baseline models, and the results on the chemical reaction classification task, where we predict the reaction type based on a given set of molecules.

### 5.1 DATASETS

We test DLGNet on common organic chemistry reaction classes, namely a variety of chemical transformations that are fundamental to both research and industrial chemistry. Those include molecular rearrangements, such as the interconversion (substitution) or the elimination of molecular substituents, as well as the introduction of specific functional groups (e.g., acyl, alkyl, or aryl groups) in a chemical compound. Other important reactions classes involve the formation of certain bond-types (e.g., carbon-carbon: C–C) or structures (e.g., heterocyclic compounds). We rely on a standard dataset (`Dataset-1`) and additionally construct two new ones (`Dataset-2` and `Dataset-3`):— see Figure 4 in Appendix D.

**Dataset-1.** As main source of data, we use the reactions from USPTO granted patents (Lugo-Martinez et al., 2021), which is the most widely used dataset for retrosynthesis problems and contains about 480K reactions. After removing duplicates and erroneous reactions, we select a subset, namely `Dataset-1`, comprising 50K atom-mapped reactions belonging to 10 different classes. An example component from `Dataset-1` is reported in Figure 4, left upper panel. The composition of the dataset is detailed in Table 3, Appendix D.

**Dataset-2.** This dataset is the result of the merging of data from five different sources and contains 5300 reactions. It presents a smaller number of reaction types, but a larger variety of substituents and reaction conditions, such as the presence of solvent or catalyst, hence providing additional complexity on some specific classes for the model to predict. Figure 4, upper right panel illustrates an example from it. Given that some elements are shared across the data sources, we combine them into three major classes. Details about `Dataset-2` composition are reported in Table 4, Appendix D.

**Dataset-3.** Since the wo datasets listed so far only include single-product reactions, in order to test the model on a highly complex task we add a third collection, `Dataset-3`, comprised of double-product bimolecular nucleophilic substitution ($S_N2$) and triple-product bimolecular elimination (E2) reaction classes, extracted from von Rudorff et al. (2020) and totaling 649 competitive reactions. A schematic representation of `Dataset-3` elements is reported in Figure 4, lower panel. Further details can be found in Appendix D.

In all three datasets, the node features are build based on *Morgan Fingerprints* (MFs) Rogers &Hahn (2010), which are one of the most widely used molecular descriptors. MFs encode a molecule by capturing the presence or absence of specific substructures (fragments) within the molecular graph. The algorithm iteratively updates the representation of each atom based on its local environment, enclosed within a radius. A radius of $r$ indicates that the environment up to $r$ bonds away from each atom is incorporated into the final representation.

### 5.2 BASELINES

We categorize our baseline models into two main groups: Hypergraph Neural Networks that handle undirected hypergraphs and those designed specifically for directed hypergraphs. Similar to Graph Neural Networks (GNNs), HNNs can be classified into spectral-based and spatial-based approaches. Spatial-based HNNs treat the convolution operator as a localized aggregation function (Dong et al., 2020). On the other hand, spectral-based HNNs define the convolution operator (based on the graph Fourier transform theory) as a function of the eigenvalue decomposition of the Laplacian matrix associated with the hypergraph Feng et al. (2019). Differently, the spectral-based one defines the convolution operator (a rigorous one in this case) as a function of the eigenvalue decomposition of the

Laplacian matrix associated with the hypergraph Feng et al. (2019). In the spectral-based category, methods such as HGNN (Feng et al., 2019), HCHA in Dong et al. (2020), HGNN$^+$ (Gao et al., 2022) are analogous to GNNs applied on clique expansions of hypergraphs. For spatial-based methods, HNNs such as HNHN (Dong et al., 2020), UniGCNII (Huang &Yang, 2021), HyperDN (Tudisco et al., 2021), LEGCN Yang et al. (2022), as well as set-based models (Chien et al., 2021), AllDeepSets and AllSetTransformer, incorporate hyperedge features and employ a message-passing framework, which can be interpreted as GNNs applied to the star expansion graph. Additionally, ED-HNN (Wang et al., 2023a) leverages gradient diffusion processes to generalize across a broad class of hypergraph neural networks, while PhenomNN (Wang et al., 2023b) introduces a framework based on hypergraph-regularized energy functions. Finally, in the context of directed HNNs, two state-of-the-art models are considered: DHM (Zhao et al., 2024) and DHRL (Ma et al., 2024). The first one, DHM, encodes high-order information in directed hypergraphs and captures the directional information of directed hyperedges through an attention mechanism and a directed hypergraph momentum encoder. The second method, DHRL, approximates the Laplacian of the directed hypergraph and formulates the convolution operation on this directed hypergraph structure.

Differently from these baselines, as shown in Section 3, DLGNet is a spectral-based GCN that leverages the Directed Line Graph Laplacian and convolves on the directed line graph derived from the directed hypergraph.

## 5.3 EXPERIMENTAL DETAILS

To solve the task of hyperedge classification, it is essential to model the molecules and their interactions within the dataset. For this reason, each molecule is represented as a node in the hypergraph, while the interactions between molecules are encoded as hyperedges. The nature of these hyperedges—whether undirected or directed—depends on the specific capabilities of the methods.

We evaluate the performance of DLGNet against these 13 state-of-the-art (baseline) methods. Since all the competitors operate directly on the undirected or directed hypergraph, we apply the feature transfer operation $X = \vec{B}^* X'$ described in Section 3 (more details in Appendix E) after the convolutional layers. After this step, each method is equipped with $\ell$ linear layers. The hyperparameters of these baselines and of our proposed model are selected via grid search (see Appendix E). The datasets are split into 50% for training, 25% for validation, and 25% for testing. The experiments are conducted with 5 random data splits, reporting the average F1-score across the splits. We choose the F1-score as evaluation metric due to the class imbalance naturally present in the datasets. Throughout the tables contained in this section, the best results are reported in **boldface** and the second best are underlined. The datasets and code we used are available on GitHub (see Appendix A).

## 5.4 RESULTS

**Quantitative.** The F1-score along with the relative standard deviation across different methods, datasets, and folds is presented in Table 1. The results show that, across the three datasets, DLGNet achieves an average additive performance improvement over the best-performing competitor of approximately 3.35 percentage points. In terms of Relative Percentage Difference (RPD)[3], we have an average RPD improvement of 3.27%. DLGNet achieves the best improvement on `Dataset-2`, with a RPD improvement of approximately 5.28% and an average additive improvement of 5.43 percentage points over the second-best competitor.

A clear trend emerges: HNN-based methods for undirected hypergraphs consistently underperform compared to those designed for directed hypergraphs, such as DHM and DHRL. Notably, our proposed DLGNet, operating on the directed line graph, outperforms all competitors, including DHM and DHRL.

**Qualitative.** To gain deeper insights into the capability of DLGNet of classifying different reaction types, we analyze the confusion matrices for `Dataset-1` and `Dataset-2`. The results of this analysis are presented in Figure 5 and Figure 6 in Appendix F. The confusion matrix for `Dataset-1` reveals that, while most of the classes are predicted extremely well, e.g., Protection and Functional

---

[3]The RPD of two values $P_1, P_2$ is the percentage ratio of their difference to their average, i.e., $|P_1 - P_2|/\frac{P_1+P_2}{2}\%$.

Table 1: Mean F1-score and standard deviation obtained on the hyperedge classification task.

| Topology | Method | Dataset-1 | Dataset-2 | Dataset-3 |
|---|---|---|---|---|
| Hypergraph | HGNN | $9.71 \pm 3.02$ | $36.40 \pm 7.27$ | $64.97 \pm 1.36$ |
| | HCHA/HGNN$^+$ | $9.99 \pm 1.91$ | $39.89 \pm 4.93$ | $63.46 \pm 2.58$ |
| | HCHA w/ Attention | $9.90 \pm 2.25$ | $11.32 \pm 0.16$ | $35.55 \pm 1.31$ |
| | HNHN | $6.95 \pm 0.97$ | $25.04 \pm 3.45$ | $52.97 \pm 5.17$ |
| | UniGCNII | $8.20 \pm 2.39$ | $29.86 \pm 0.31$ | $50.97 \pm 6.84$ |
| | HyperND | $4.63 \pm 0.04$ | $28.98 \pm 0.46$ | $52.71 \pm 12.32$ |
| | AllDeepSets | $7.64 \pm 2.23$ | $30.45 \pm 0.27$ | $51.72 \pm 5.99$ |
| | AllSetTransformer | $8.63 \pm 2.62$ | $30.67 \pm 0.57$ | $49.24 \pm 3.98$ |
| | ED-HNN | $9.19 \pm 1.43$ | $30.47 \pm 0.56$ | $50.52 \pm 10.17$ |
| | PhenomNN | $8.33 \pm 2.77$ | $29.43 \pm 0.39$ | $51.82 \pm 9.33$ |
| Directed Hypergraph | DHM | $46.04 \pm 0.58$ | $59.31 \pm 4.04$ | $68.10 \pm 3.60$ |
| | DHRL | $58.15 \pm 1.58$ | $79.36 \pm 3.94$ | $99.27 \pm 0.79$ |
| Directed-Line Graph | **DLGNet** | $\mathbf{60.55 \pm 0.80}$ | $\mathbf{83.67 \pm 3.41}$ | $\mathbf{99.75 \pm 0.34}$ |

group addition reactions (accuracy of 88% and 77%, respectively), some are predicted not as well, e.g., Functional group interconversion (41%). To better understand this behavior, we conducted a thorough inspection of the structural features of Dataset-1's components, selecting several elements from pairs of classes among which the model yields the highest uncertainty. Two example cases are reported in Figure 3. Overall, our analysis reveals that the pair of classes which are subject to the higher degree of confusion are, structurally, highly similar, which well explains the poorer performance that DLGNet achieves on them, as we illustrate in the following. The left panel illustrates the mislabeling of *Class 9* (Functional group interconversions, correctly predicted in 41% of the cases) with *Class 7* (Reductions, incorrectly predicted in 14% of the cases), while the right panel presents an example of *Class 4* (Heterocycle formations, correctly predicted in 44% of the cases) with *Class 1* (Arylations, incorrectly predicted 30% of the cases). Notably, in these examples, both the main backbone structure of the molecules and the substituent groups (the segments affected by the reactive process, highlighted in the figure) exhibit a high degree of similarity between the two classes. In the left panel, the reactants of both classes present a 6-carbon ring (in grey) as well as a iodine substituent (in purple). The atoms composing the highlighted groups are also of the same types. On the other hand, in the right panel, the majority of the constituent parts of the products are in common between the two classes. Specifically, despite the outcome of *Class 4* is the formation of a heterocycle, i.e., a hexagonal ring containing a heteroatom (nitrogen, in blue), such a geometrical feature is also present in *Class 1* arylation product, as the resulting molecule presents two heterocycles rings. Similar considerations apply to the incorrect labeling of Dataset-2 N-arylation sub classes, where the main difference between the reactants lies in the nature of the aryl halide that participates in the coupling reaction. In summary, we conclude that the model demonstrates strong predictive performance across the majority of the classes, although a few, particularly those with shared elements, remain challenging to differentiate. Nevertheless, we are confident that DLGNet will prove highly valuable to the chemistry community, allowing for the categorization of existing data sources as well as for planning new synthetic routes.

**Ablation study.** Table 2 presents the results of an ablation study carried out on DLGNet to assess the importance of directionality in DLGNet's line graph. To do this, we test DLGNet using an undirected line graph and demonstrate that DLGNet consistently outperforms its undirected counterpart on all three data sets. This indicates that directionality plays a crucial role in solving the chemical reaction classification task. Focusing on equation 12, we test DLGNet under two conditions: *i)* using $\vec{\mathbb{Q}}_N$ instead of $\vec{\mathbb{L}}_N$, and *ii)* setting $\Theta_0 = 0$, thus nullifying the first term in equation 12. The first comparison shows identical results across all datasets, thus providing a computational confirmation of the results of Proposition 1, while the restricted version of DLGNet with $\Theta_0 = 0$ performs worse. Finally, we assess the architectural choice related to the incorporation of skip connections. While DLGNet without skip connections exhibits a slight drop in performance, the results remain close to those of the original architecture.

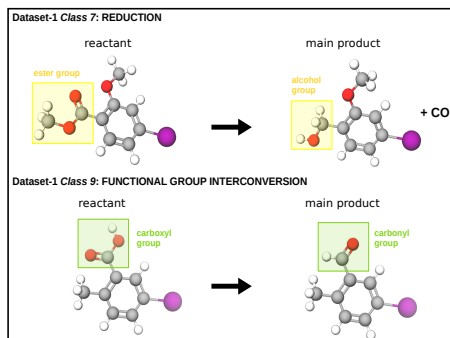
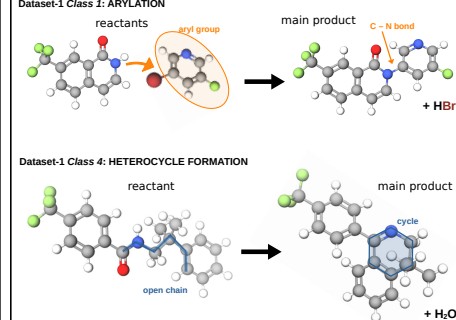

Figure 3: Ball-and-stick 3D model of `Dataset-1` mislabeled pairs of reaction classes. Color code: grey for carbon, red for oxygen, blue for nitrogen, purple for iodine, green for chlorine, light green for fluorine, brown for bromide, and white for hydrogen. **(Left panel, upper)**: Reduction from a ester to a alcohol substituent on a 6-carbon atoms ring. **(Left panel, lower)**: Functional group interconversion from carboxyl to carbonyl group in the analog hexagonal structure. **(Right panel, upper)**: arylation reaction between a amine compound and a aryl halide, yielding a C–N bond in the final product. **(Right panel, lower)**: heterocycle formation via amide intramolecular condensation, producing a hexagonal ring containing a heteroatom (nitrogen).

Table 2: Ablation study. Average F1-score and standard deviation are reported.

| Method | Dataset-1 | Dataset-2 | Dataset-3 |
|---|---|---|---|
| **DLGNet** | **60.55 ± 0.80** | **83.67 ± 3.41** | **99.75 ± 0.34** |
| DLGNet w/o directionality | 52.07 ± 1.61 | 70.19 ± 0.65 | 81.65 ± 8.39 |
| DLGNet w/ Signless Laplacian | 60.24 ± 0.36 | 82.86 ± 1.96 | **99.75 ± 0.55** |
| DLGNet w/ $\Theta_0 = 0$ | 53.82 ± 0.74 | 75.68 ± 3.59 | 91.45 ± 2.36 |
| DLGNet w/o skip-connection | 56.38 ± 3.02 | 80.63 ± 3.54 | 99.63 ± 0.34 |

# 6   CONCLUSIONS

We introduced the Directed Line Graph Network (DLGNet), the first spectral GNN specifically designed to operate on directed line graphs associated with directed hypergraphs by directly convolving hyperedge features. DLGNet leverages a novel complex-valued Laplacian matrix, the *Directed Line Graph Laplacian*, which is a Hermitian matrix encoding the interactions among the hyperedges of a hypergraph using complex numbers. This formulation allows for the natural representation of both directed and undirected relationships between the hyperedges, capturing rich structural information. Our proposed DLGNet network utilizes this new Laplacian matrix to perform spectral convolutions on the line graph featuring both undirected and directed edges. Via the Directed Line Graph representation, our proposed model enables the seamless integration of the directionality present in the hypergraph at hand, which is crucial for accurately modeling various real-world phenomena involving asymmetric high-order interactions.

We evaluated our approach on the chemical reaction classification problem using three real-world datasets. In these experiments, we demonstrated the superiority of DLGNet, which achieved an average relative percentage difference improvement of 3.27% over the second-best method across the three datasets. This highlights the importance of directly convolving the hypergraph features on the directed line graph, instead of doing so in the undirecetd/directed hypergraph. Though an ablation study, we demonstrated the relevance of encoding directional information via the directed line graph associated with a directed hypergraph as opposed to ignoring it. We also provided a qualitative analysis DLGNet's results in light of the underlying chemical reaction classification task.

In light of the promising results we obtained and as a future perspective, we would like to address more complex and challenging tasks, such as retrosynthetic planning and reaction discovery, which require sophisticated analysis and deeper insights into the underlying chemical processes.

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
