## A    CODE REPOSITORY AND LICENSING

The code developed for this research work is available at `https://anonymous.4open.science/r/HyperedgeClassification-2E63` and freely distributed under the Apache 2.0 license.[4] code for the baselines used in the experimental analysis is available at `https://github.com/Graph-COM/ED-HNN`, `https://github.com/yxzwang/PhenomNN` and `https://github.com/WBZhao98/DHMConv` under the MIT license.[5]

## B    PROPERTIES OF OUR PROPOSED LAPLACIAN

This section contains the proofs of the theorems and corollaries reported in the main paper.

**Theorem 1.** *If $\vec{H}$ is undirected (i.e., $\vec{H} = H$), $\vec{\mathbb{L}}_N = \mathbb{L}_N$ and $\vec{\mathbb{Q}}_N = \mathbb{Q}_N$ holds.*

*Proof.* Since $H = (V, E)$ is an undirected hypergraph, $\vec{B}$ is binary and only takes values 0 and 1 (rather than being ternary and taking values $0, 1, -i$), defining an undirected line graph $L(H)$. In particular, for each edge $e \in E$ we have $\vec{B}_{ue} = 1$ if either $u \in H(e)$ or $u \in T(e)$ and $\vec{B}_{ue} = 0$ otherwise. Consequently, the directed incident matrix $\vec{B}$ is identical to the non-directed incidence matrix $B$, i.e., $\vec{B} = B$. Thus, by construction, $\vec{\mathbb{L}}_N = \mathbb{L}_N$ and $\vec{\mathbb{Q}}_N = \mathbb{Q}_N$.    □

**Theorem 2.** *Letting $\mathbf{1}$ be the indicator function, the Euclidean norm induced by $\vec{\mathbb{L}}_N$ of a complex-valued signal $x = a + ib \in \mathbb{C}^m$ with a component per hyperedge in $E$ reads:*

$$
\frac{1}{2} \sum_{u \in V} \frac{1}{d(u)} \sum_{i,j \in E} \left( \left( \frac{w(j)^{\frac{1}{2}} a_i}{\delta(i)^{\frac{1}{2}}} - \frac{w(i)^{\frac{1}{2}} a_j}{\delta(j)^{\frac{1}{2}}} \right)^2 + \left( \frac{w(j)^{\frac{1}{2}} b_i}{\delta(i)^{\frac{1}{2}}} - \frac{w(i)^{\frac{1}{2}} b_j}{\delta(j)^{\frac{1}{2}}} \right)^2 \right) \mathbf{1}_{u \in H(i) \cap H(j) \vee u \in T(i) \cap T(j)}
$$

$$
+ \left( \left( \frac{w(j)^{\frac{1}{2}} a_i}{\delta(i)^{\frac{1}{2}}} - \frac{w(i)^{\frac{1}{2}} b_j}{\delta(j)^{\frac{1}{2}}} \right)^2 + \left( \frac{w(i)^{\frac{1}{2}} a_j}{\delta(j)^{\frac{1}{2}}} + \frac{w(j)^{\frac{1}{2}} b_i}{\delta(i)^{\frac{1}{2}}} \right)^2 \right) \mathbf{1}_{u \in H(i) \cap T(j)}
$$

$$
+ \left( \left( \frac{w(j)^{\frac{1}{2}} a_i}{\delta(i)^{\frac{1}{2}}} + \frac{w(i)^{\frac{1}{2}} b_j}{\delta(j)^{\frac{1}{2}}} \right)^2 + \left( \frac{w(i)^{\frac{1}{2}} a_j}{\delta(j)^{\frac{1}{2}}} - \frac{w(j)^{\frac{1}{2}} b_i}{\delta(i)^{\frac{1}{2}}} \right)^2 \right) \mathbf{1}_{u \in T(i) \cap H(j)}. \tag{13}
$$

*Proof.* By definition, we have

$$
x^* \vec{\mathbb{L}}_N x = x^* I x - x^* \vec{\mathbb{Q}}_N x = x^* I x - x^* \vec{D}_e^{-\frac{1}{2}} W^{\frac{1}{2}} \vec{B}^* \vec{D}_v^{-1} \vec{B} W^{\frac{1}{2}} \vec{D}_e^{-\frac{1}{2}}.
$$

Scalarly, the expression reads

$$
\sum_{i \in E} x_i^* x_i - \sum_{i,j \in E} \sum_{u \in V} \frac{1}{d(u)} \frac{w(i)^{\frac{1}{2}} \vec{B}(u,i)^* \vec{B}(u,j) w(j)^{\frac{1}{2}}}{\delta(i)^{\frac{1}{2}} \delta(j)^{\frac{1}{2}}} x_i^* x_j,
$$

where $\sum_{i,j \in E}$ indicates the sum over all ordered pairs $i, j$ in $E$, including those where $i = j$. W.l.o.g., we can swap the order of the sums in the second term, obtaining:

$$
\sum_{i \in E} x_i^* x_i - \sum_{u \in V} \sum_{i,j \in E} \frac{1}{d(u)} \frac{w(i)^{\frac{1}{2}} \vec{B}(u,i)^* \vec{B}(u,j) w(j)^{\frac{1}{2}}}{\delta(i)^{\frac{1}{2}} \delta(j)^{\frac{1}{2}}} x_i^* x_j.
$$

Due to $\mathbb{Q}_N$ being Hermitian, $\mathbb{Q}_N + \mathbb{Q}_N^* = 2\mathbb{Q}_N$ holds. Thus, substituting $\frac{1}{2}(\mathbb{Q}_N + \mathbb{Q}_N^*)$ for $\mathbb{Q}_N$, we can rewrite the second term as

$$
-\frac{1}{2} \sum_{u \in V} \frac{1}{d(u)} \sum_{i,j \in E} w(i)^{\frac{1}{2}} \left( \vec{B}(u,i)^* \vec{B}(u,j) \frac{x_i^* x_j}{\delta(i)^{\frac{1}{2}} \delta(j)^{\frac{1}{2}}} + \vec{B}(u,j)^* \vec{B}(u,i) \frac{x_j^* x_i}{\delta(j)^{\frac{1}{2}} \delta(i)^{\frac{1}{2}}} \right) w(j)^{\frac{1}{2}}.
$$

Next, we show that the following holds for the first term:

$$
\sum_{i \in E} x_i^* x_i = \sum_{u \in V} \frac{1}{d(u)} \sum_{i,j \in E : u \in i \wedge u \in j} w(j) \frac{x_i^* x_i}{\delta(i)}.
$$

---

[4]`https://www.apache.org/licenses/LICENSE-2.0`
[5]`https://choosealicense.com/licenses/mit/`

We show this by showing how to turn the right-hand side into the left-hand side. First, we pre-pone the sum over $i$ in the right-hand side, obtaining:

$$\sum_{i \in E} \left( \sum_{u \in V} \frac{1}{d(u)} \sum_{j \in E : u \in j} w(j) \frac{x_i^* x_i}{\delta(i)} \right).$$

Then, we bring $\frac{1}{\delta(i)}$ and $x_i^* x_i$ outside of the inner summation, which leads to the following expression

$$= \sum_{i \in E} x_i^* x_i \frac{1}{\delta(i)} \underbrace{\sum_{u \in V} \underbrace{\frac{1}{d(u)} \sum_{j \in E : u \in j} w(j)}_{=1}}_{=1}.$$

Following the calculations reported as underbraces, we deduce that the coefficient that multiplies $x_i^* x_j$ is equal to 1, concluding this part of the proof.

As we did for the second term, we now double the summation in the first term and compensate for it with a factor of $\frac{1}{2}$, obtaining:

$$\frac{1}{2} \sum_{u \in V} \frac{1}{d(u)} \sum_{i,j \in E : u \in i \wedge u \in j} \left( w(j) \frac{x_i^* x_i}{\delta(i)} + w(i) \frac{x_j^* x_j}{\delta(j)} \right).$$

Looking back at both terms, we have at the expression

$$\frac{1}{2} \sum_{u \in V} \frac{1}{d(u)} \sum_{i,j \in E : u \in i \wedge u \in j} \left( w(j) \frac{x_i^* x_i}{\delta(i)} + w(i) \frac{x_j^* x_j}{\delta(j)} \right) +$$

$$- \frac{1}{2} \sum_{u \in V} \frac{1}{d(u)} \sum_{i,j \in E} w(i)^{\frac{1}{2}} \left( \vec{B}(u,i)^* \vec{B}(u,j) \frac{x_i^* x_j}{\delta(i)^{\frac{1}{2}} \delta(j)^{\frac{1}{2}}} + \vec{B}(u,j)^* \vec{B}(u,i) \frac{x_j^* x_i}{\delta(j)^{\frac{1}{2}} \delta(i)^{\frac{1}{2}}} \right) w(j)^{\frac{1}{2}}.$$

After rewriting the second summation in the second term as $\sum_{i,j \in E : u \in i \wedge j}$ (this is w.l.o.g. due to the summand being 0 if either $u \notin i$ or $u \notin i$), we compactly rewrite the whole expression as

$$\frac{1}{2} \sum_{u \in V} \frac{1}{d(u)} \sum_{i,j \in E : u \in i \wedge u \in j} \left( w(j) \frac{x_i^* x_i}{\delta(i)} + w(i) \frac{x_j^* x_j}{\delta(j)} + \right.$$

$$\left. - w(i)^{\frac{1}{2}} w(j)^{\frac{1}{2}} \vec{B}(u,i)^* \vec{B}(u,j) \frac{x_i^* x_j}{\delta(i)^{\frac{1}{2}} \delta(j)^{\frac{1}{2}}} - w(i)^{\frac{1}{2}} w(j)^{\frac{1}{2}} \vec{B}(u,j)^* \vec{B}(u,i) \frac{x_j^* x_i}{\delta(j)^{\frac{1}{2}} \delta(i)^{\frac{1}{2}}} \right).$$

Now, we proceed by analyzing the three possible cases for the summand.

Case 1.a: $u \in H(i) \cap H(j) \Leftrightarrow \vec{B}(u,i) = 1, \vec{B}(u,j) = 1$. We have $\vec{B}(u,i)^* \vec{B}(u,j) = \vec{B}(u,j)^* \vec{B}(u,i) = 1$.

Case 1.b: $u \in T(i) \cap T(j) \Leftrightarrow \vec{B}(u,i) = -\mathrm{i}, \vec{B}(u,j) = -\mathrm{i}$. We have $\vec{B}(u,i)^* \vec{B}(u,j) = \vec{B}(u,j)^* \vec{B}(u,i) = (-\mathrm{i})^* (-\mathrm{i}) = (-\mathrm{i})(\mathrm{i}) = 1$.

In both cases, we have:

$$w(j) \frac{x_i^* x_i}{\delta(i)} + w(i) \frac{x_j^* x_j}{\delta(j)} - w(i)^{\frac{1}{2}} w(j)^{\frac{1}{2}} \frac{x_i^* x_j}{\delta(i)^{\frac{1}{2}} \delta(j)^{\frac{1}{2}}} - w(i)^{\frac{1}{2}} w(j)^{\frac{1}{2}} \frac{x_j^* x_i}{\delta(j)^{\frac{1}{2}} \delta(i)^{\frac{1}{2}}} =$$

$$\left( \frac{w(j)^{\frac{1}{2}} x_i}{\delta(i)^{\frac{1}{2}}} - \frac{w(i)^{\frac{1}{2}} x_j}{\delta(j)^{\frac{1}{2}}} \right)^* \left( \frac{w(j)^{\frac{1}{2}} x_i}{\delta(i)^{\frac{1}{2}}} - \frac{w(i)^{\frac{1}{2}} x_j}{\delta(j)^{\frac{1}{2}}} \right).$$

Letting $x_i = a_i + \mathrm{i} b_i$ and $x_j = a_j + \mathrm{i} b_j$, this expression boils down to

$$\left( \frac{w(j)^{\frac{1}{2}} a_i}{\delta(i)^{\frac{1}{2}}} - \frac{w(i)^{\frac{1}{2}} a_j}{\delta(j)^{\frac{1}{2}}} \right)^2 + \left( \frac{w(j)^{\frac{1}{2}} b_i}{\delta(i)^{\frac{1}{2}}} - \frac{w(i)^{\frac{1}{2}} b_j}{\delta(j)^{\frac{1}{2}}} \right)^2.$$

Case 2.a: $u \in H(i) \cap T(j) \Leftrightarrow \bar{B}(u,i) = 1, \bar{B}(u,j) = -\mathrm{i}$. We have $\bar{B}(u,i)^* \bar{B}(u,j) = (1)^*(-\mathrm{i}) = -\mathrm{i}$ and $\bar{B}(u,j)^* \bar{B}(u,i) = (-\mathrm{i})^*(1) = \mathrm{i}$. In this case, we have:

$$w(j)\frac{x_i^* x_i}{\delta(i)} + w(i)\frac{x_j^* x_j}{\delta(j)} + \mathrm{i}w(i)^{\frac{1}{2}}w(j)^{\frac{1}{2}}\frac{x_i^* x_j}{\delta(i)^{\frac{1}{2}}\delta(j)^{\frac{1}{2}}} - \mathrm{i}w(i)^{\frac{1}{2}}w(j)^{\frac{1}{2}}\frac{x_j^* x_i}{\delta(j)^{\frac{1}{2}}\delta(i)^{\frac{1}{2}}}.$$

Letting $x_i = a_i + \mathrm{i}b_i$ and $x_j = a_j + \mathrm{i}b_j$, this expression reads

$$\left(\frac{w(j)^{\frac{1}{2}}a_i}{\delta(i)^{\frac{1}{2}}} - \frac{w(i)^{\frac{1}{2}}b_j}{\delta(j)^{\frac{1}{2}}}\right)^2 + \left(\frac{w(i)^{\frac{1}{2}}a_j}{\delta(j)^{\frac{1}{2}}} + \frac{w(j)^{\frac{1}{2}}b_i}{\delta(i)^{\frac{1}{2}}}\right)^2.$$

Case 2.b: $u \in T(i) \cap H(j) \Leftrightarrow \bar{B}(u,i) = -\mathrm{i}, \bar{B}(u,j) = 1$. We have $\bar{B}(u,i)^* \bar{B}(u,j) = (-\mathrm{i})^*(1) = \mathrm{i}$ and $\bar{B}(u,j)^* \bar{B}(u,i) = (1)^*(-\mathrm{i}) = -\mathrm{i}$. In this case, we have:

$$w(j)\frac{x_i^* x_i}{\delta(i)} + w(i)\frac{x_j^* x_j}{\delta(j)} - \mathrm{i}w(i)^{\frac{1}{2}}w(j)^{\frac{1}{2}}\frac{x_i^* x_j}{\delta(i)^{\frac{1}{2}}\delta(j)^{\frac{1}{2}}} + \mathrm{i}w(i)^{\frac{1}{2}}w(j)^{\frac{1}{2}}\frac{x_j^* x_i}{\delta(j)^{\frac{1}{2}}\delta(i)^{\frac{1}{2}}}.$$

Letting $x_i = a_i + \mathrm{i}b_i$ and $x_j = a_j + \mathrm{i}b_j$, this latter expression reads

$$\left(\frac{w(j)^{\frac{1}{2}}a_i}{\delta(i)^{\frac{1}{2}}} + \frac{w(i)^{\frac{1}{2}}b_j}{\delta(j)^{\frac{1}{2}}}\right)^2 + \left(\frac{w(i)^{\frac{1}{2}}a_j}{\delta(j)^{\frac{1}{2}}} - \frac{w(j)^{\frac{1}{2}}b_i}{\delta(i)^{\frac{1}{2}}}\right)^2.$$

The final equation reported in the statement of the theorem is obtained by combining the four cases we just analyzed. $\qquad\square$

**Corollary 1.** $\vec{\mathbb{L}}_N$ *is positive semidefinite.*

*Proof.* Since $\vec{\mathbb{L}}_N$ is Hermitian, it can be diagonalized as $U\Lambda U^*$ for some $U \in \mathbb{C}^{n \times n}$ and $\Lambda \in \mathbb{R}^{n \times n}$, where $\Lambda$ is diagonal and real. We have $x^*\vec{\mathbb{L}}_N x = x^* U\Lambda U^* x = y^*\Lambda y$ with $y = U^* x$. Since $\Lambda$ is diagonal, we have $y^*\Lambda y = \sum_{u \in V} \lambda_u y_u^2$. Thanks to Theorem 2, the quadratic form $x^*\vec{\mathbb{L}}_N x$ associated with $\vec{\mathbb{L}}_N$ is a sum of squares of real values and, hence, nonnegative. Combined with $x^*\vec{\mathbb{L}}_N x = \sum_{u \in L(V)} \lambda_u y_u^2$, we deduce $\lambda_u \geq 0$ for all $u \in L(V)$, where $L(V)$ is the vertex set of DLG($\vec{H}$). $\qquad\square$

**Corollary 2.** $\lambda_{\max}(\vec{\mathbb{L}}_N) \leq 1$ *and* $\lambda_{\max}(\vec{\mathbb{Q}}_N) \leq 1$.

*Proof.* $\lambda_{\max}(\vec{\mathbb{L}}_N) \leq 1$ holds if and only if $\vec{\mathbb{L}}_N - I \preceq 0$. Since $\vec{\mathbb{L}}_N = I - \vec{\mathbb{Q}}_N$ holds by definition, we need to prove $-\vec{\mathbb{Q}}_N \preceq 0$. This is the case due to Theorem 3.

Similarly, $\lambda_{\max}(\vec{\mathbb{Q}}_N) \leq 1$ holds if and only if $\vec{\mathbb{Q}}_N - I \preceq 0$. Since $\vec{\mathbb{Q}}_N = I - \vec{\mathbb{L}}_N$ holds by definition, we need to prove $-\vec{\mathbb{L}}_N \preceq 0$. This is the case due to Theorem 1. $\qquad\square$

**Directed Line Graph Laplacian and The Other Laplacians** Examining the behavior of the Directed Line Graph Laplacian through Equation 10, we observe that it differs from other Laplacians designed to handle both directed and undirected edges in graphs, such as the *Magnetic Laplacian* (Lieb &Loss, 1993) and the *Sign Magnetic Laplacian* (Fiorini et al., 2023). Indeed, the Directed Line Graph Laplacian exhibits a unique characteristic: both its real and imaginary components can be simultaneously non-zero. This is different from the case of the *Sign Magnetic Laplacian*, which can only have one of the two components different from zero at any given time, and also from the case of the *Magnetic Laplacian*, which coincides with the *Sign Magnetic Laplacian* when $q = \frac{1}{4}$ and the graph has binary weights. Let us note that the *Magnetic Laplacian* can also have both components different from zero, but such a behavior is influenced by both the edge weight and the value of $q$, and may lead to the sign-pattern inconsistency described in Fiorini et al. (2023), which our proposed Directed Line Graph Laplacian does not suffer from.

## C  COMPLEXITY OF DLGNET

The detailed calculations for the (inference) complexity of DLGNet are as follows.

1. The Directed Line Graph Laplacian $\vec{\mathbb{L}}_N$ is constructed in time $O(m^2 n)$, where the factor $n$ is due to the need for computing the product between two columns of $\vec{B}$ (i.e., two rows of $B^*$) to calculate each entry of $\vec{\mathbb{L}}_N$. After $\vec{\mathbb{L}}_\mathbb{N}$ has been computed, the convolution matrix $\hat{Y} \in \mathbb{C}^{m \times m}$ is constructed in time $O(m^2)$. Note that such a construction is carried out entirely in pre-processing and is not required at inference time.

2. Constructing the feature matrix $X = \vec{B}^* X'$ requires $O(mnc_0)$ elementary operations.

3. Each of the $\ell$ convolutional layers of DLGNet requires $O(m^2 c + mc^2 + mc) = O(m^2 c + mc^2)$ elementary operations across 3 steps. Let $X^{l-1}$ be the input matrix to layer $l = 1, \ldots, \ell$. The operations that are carried out are the following ones.

   (a) $\vec{\mathbb{L}}_\mathbb{N}$ is multiplied by the hyperedge-feature matrix $X^{l-1} \in \mathbb{C}^{m \times c}$, obtaining $P^{l_1} \in \mathbb{C}^{m \times c}$ in time $O(m^2 c)$ (we assume, for simplicity, that matrix multiplications takes cubic time);

   (b) The matrices $P^{l_0} = IX^{l-1} = X^{l-1}$ and $P^{l_1}$ are multiplied by the weight matrices $\Theta_0, \Theta_1 \in \mathbb{R}^{c \times c}$ (respectively), obtaining the intermediate matrices $P^{l_{01}}, P^{l_{11}} \in \mathbb{C}^{n \times c}$ in time $O(mc^2)$.

   (c) The matrices $P^{l_{01}}$ and $P^{l_{11}}$ are additioned in time $O(mc)$ to obtain $P^{l_2}$.

   (d) The activation function $\phi$ is applied component wise to $P^{l_2}$ in time $O(mc)$, resulting in the output matrix $X^l \in \mathbb{C}^{m \times c}$ of the $l$-th convolutional layer.

4. The unwind operator transforms $X^\ell$ (the output of the last convolutional layer $\ell$) into the matrix $U^0 \in \mathbb{R}^{n \times 2c}$ in linear time $O(mc)$.

5. Call $U^{s-1}$ the input matrix to each linear layer of index $s = 1, \ldots, S$. The application of the $s$-th linear layer to $U^{s-1} \in \mathbb{C}^{m \times c'}$ requires multiplying $U^{s-1}$ by a weight matrix $M_s \in \mathbb{C}^{c' \times c'}$ (where $c'$ is the number of channels from which and into which the feature vector of each node is projected). This is done in time $O(mc'^2)$.

6. In the last linear layer of index $S$, the input matrix $U^{S-1} \in \mathbb{R}^{m \times c'}$ is projected into the output matrix $O \in \mathbb{R}^{m \times d}$ in time $O(nc'd)$.

7. The application of the Softmax activation function takes linear time $O(md)$.

We deduce an overall complexity of $O(mnc_0) + O(\ell(m^2 c + mc^2) + mc + (S-1)(mc'^2) + mc'd + md)$. Assuming $O(c) = O(c') = O(d) = \bar{c}$, such a complexity coincides with $O(\ell(m^2 \bar{c}) + (\ell + S)(m\bar{c}^2))$.

## D  FURTHER DETAILS ON THE DATASETS

Details on the datasets composition are reported in Tables 3, 4, 5. Most of the elements of `Dataset-1` belong to the first two classes, which concern the addition of functional groups to a chemical compound: alkyl and aryl groups for *Class 1* and acyl groups for *Class 2*, comprising more than 17K species. Less populated classes involve specific chemical transformations, such as *Class 3* (C–C bond formation) which contains less than 1000 elements.

`Dataset-2` presents solely three classes. The elements of the first class (C–C bond formation) are extracted from two separated collections present in the Open Reaction Database (ORD) Project (Kearnes et al., 2021). Those are the Reizman et al. (2016) data for the Pd-catalyzed Suzuki–Miyaura cross-coupling reactions and a vast collection of Pd-catalyzed imidazole-aryl coupling reactions, via C-H arylation. The elements of *Class 2* (N-arylation) includes data of Pd-catalyzed N-arylation (Buchwald-Hartwig) reactions from the AstraZeneca ELN dataset, also generated from the ORD website. This class has been further divided in 3 sub-classes according to the nature of the aryl halide used for the coupling. Finally, the third class contains an ORD collection of data for amide bond formation processes. We have been able to extract sub-categories from two of them. Those are *Class 1* (C–C bond formation) and *Class 2* (N-arylation processes) and contain two and three sub-classes, respectively. The most populated class is Imidazole-aryl coupling, comprising

around 1500 elements belonging to the class of palladium-catalyzed imidazole C-H arylation. The chemical diversity in this class is ensured by the use of 8 aryl bromides and 8 imidazole compunds. Furthermore, in terms of reaction conditions, the collection presents 24 different monophosphine ligands.

Unlike the previous ones, `Dataset-3` has been assembled starting from competitive processes; therefore it contains almost the same amount of elements ($\sim 300$) for the two classes: Bimolecular nucleophilic substitution ($S_N2$) and eliminations (E2). The reactants–which are in common between $S_N2$ and E2—are substituted alkane compounds and nucleophile agents. The substituents span a range of electron donating and electron withdrawing effect strengths, including methyl, cyano, amine, and nitro functional groups. The nucleophiles have been chosen either between halide or hydrogen anions, while the molecular skeleton is ethane.

Table 3: Distribution of the reactions in the Dataset-1.

| Rxn class | Rxn name | Num rxns |
|---|---|---|
| 1 | Heteroatom alkylation and arylation | 15151 |
| 2 | Acylation and related process | 11896 |
| 3 | C-C bond formation | 909 |
| 4 | Heterocycle formation | 4614 |
| 5 | Protections | 1834 |
| 6 | Deprotections | 5662 |
| 7 | Reductions | 672 |
| 8 | Oxidations | 811 |
| 9 | Functional group interconversion | 8237 |
| 10 | Functional group addition | 230 |

Table 4: Distribution of the reactions in the Dataset-2.

| Rxn class | Rxn name | Num rxns |
|---|---|---|
| 1 | C-C bond formation | 1921 |
| | - Reizman Suzuki Cross-Coupling | 385 |
| | - Imidazole-aryl coupling | 1536 |
| 2 | Heteroatom (N) arylation: | 657 |
| | - Amine + Aryl bromide | 278 |
| | - Amine + Aryl chloride | 299 |
| | - Amine + Aryl iodide | 80 |
| 3 | Amide bond formation | 960 |

Table 5: Distribution of reactions in the Dataset-3.

| Rxn class | Rxn name | Num rxns |
|---|---|---|
| 1 | Bimolecular nucleophilic substitution ($S_N2$) | 301 |
| 2 | Bimolecular elimination (E2) | 348 |

## E  FURTHER DETAILS ON THE EXPERIMENTS

**Hardware.**   The experiments were conducted on 2 different machines:

1. An Intel(R) Xeon(R) Gold 6326 CPU @ 2.90GHz with 380 GB RAM, equipped with an NVIDIA Ampere A100 40GB.

2. A 12th Gen Intel(R) Core(TM) i9-12900KF CPU @ 3.20GHz CPU with 64 GB RAM, equipped with an NVIDIA RTX 4090 GPU.

**Model Settings.**   We trained every learning model considered in this paper for up to 1000 epochs. We adopted a learning rate of $5 \cdot 10^{-3}$ and employed the optimization algorithm Adam with weight

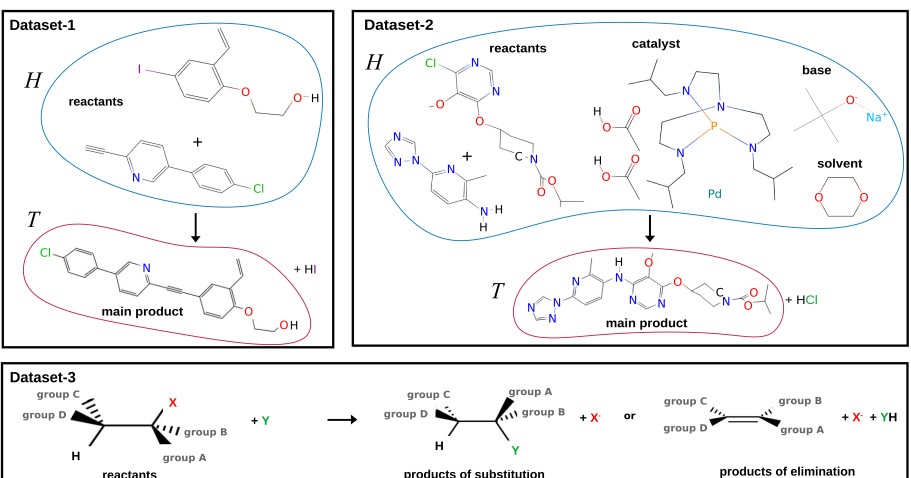

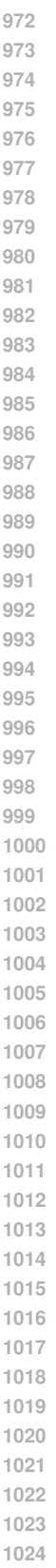

Figure 4: **(Upper panel, left)**: example from `Dataset-1`. C–C bond formation via reaction of alkyne with alkyl halide; only bi-molecular reactant and main product are taken into account (any byproduct is omitted). **(Upper panel, right)**: example from `Dataset-2`. C–N bond formation via Buchwald-Hartwig amination; apart from bi-molecular reactant (amine and aryl halide) and main product, catalyst (palladium compound), solvent (dioxane) and base (sodium tert-butoxide) structures are also present. Chemical elements: carbon (C), nitrogen (N), oxygen (O), hydrogen (H), chlorine (Cl), iodine (I), sodium (Na), phosphorus (P) and palladium (Pd). Single, double and triple black lines: bonds between C atoms. $H$, $T$: Head and Tail of the directed hypergraph. **(Lower panel)**: schematic representation of `Dataset-3` elements. Left side: reactants; right side: competitive outcomes between bimolecular nucleophilic substitution ($S_N2$) or bimolecular elimination (E2). Thus, each element is composed either of a bi-molecular reactant and a bi-molecular product ($S_N2$ class), or a bi-molecular reactant and a tri-molecular product (E2 class). X and Y: leaving group and nucleophile agent. Groups A-D: different substituents attached to the alkane carbon backbone (black).

decays equal to $5 \cdot 10^{-4}$ (in order to avoid overfitting). We set the number of linear layers to 2, i.e. $\ell = 2$, for all the models.

We adopted a hyperparameter optimization procedure to identify the best set of parameters for each model. In particular, the hyperparameter values are:

- For AllDeepSets and ED-HNN, the number of basic block is chosen in $\{1, 2, 4, 8\}$, the number of MLPs per block in $\{1, 2\}$, the dimension of the hidden MLP (i.e., the number of filters) in $\{64, 128, 256, 512\}$, and the classifier hidden dimension in $\{64, 128, 256\}$.

- For AllSetTransformer the number of basic block is chosen in $\{2, 4, 8\}$, the number of MLPs per block in $\{1, 2\}$, the dimension of the hidden MLP in $\{64, 128, 256, 512\}$, the classifier hidden dimension in $\{64, 128, 256\}$, and the number of heads in $\{1, 4, 8\}$.

- For UniGCNII, HGNN, HNHN, HCHA/HGNN$^+$, LEGCN, and HCHA with the attention mechanism, the number of basic blocks is chosen in $\{2, 4, 8\}$ and the hidden dimension of the MLP layer in $\{64, 128, 256, 512\}$.

- For HyperGCN, the number of basic blocks is chosen in $\{2, 4, 8\}$.

- For HyperND, the classifier hidden dimension is chosen in $\{64, 128, 256\}$.

- For PhenomNN, the number of basic blocks is chosen in $\{2, 4, 8\}$. We select four different settings:

  1. $\lambda_0 = 0.1$, $\lambda_1 = 0.1$ and prop step$= 8$,
  2. $\lambda_0 = 0$, $\lambda_1 = 50$ and prop step$= 16$,
  3. $\lambda_0 = 1$, $\lambda_1 = 1$ and prop step$= 16$,
  4. $\lambda_0 = 0$, $\lambda_1 = 20$ and prop step$= 16$.

- For DHM, the number of basic blocks is chosen in $\{1, 2, 3, 4\}$ and the classifier hidden dimension is chosen in $\{64, 128, 256, 512\}$.

- For DLGNet, the number of convolutional layers is chosen in $\{1, 2, 3\}$, the number of filters in $\{64, 128, 256, 512\}$, and the classifier hidden dimension in $\{64, 128, 256\}$. We tested DLGNet both with the input feature matrix $X \in \mathbb{C}^{n \times c}$ where $\Re(X) = \Im(X) \neq 0$ and with $\Im(X) = 0$.

**How to Transfer The Features.** As mention in Section 4, a key aspect of our approach involves transferring features from the nodes of the hypergraph to their corresponding hyperedges, i.e., the nodes of the directed line graph. To clarify this mechanism, we provide a simple example. Consider a directed hypergraph $\vec{H} = (V, E)$, where the vertex set is $V = \{u, v, c\}$ and the hyperedge set consists of $E = \{e_1\}$. In $\vec{H}$, we have $H(e_1) = \{u, v\}$ and $T(e_1) = \{c\}$. Each vertex is assigned a feature vector $x'_u, x'_v, x'_c = 1$ and the hyperedge has a unit weight, i.e. $w_{e_1} = 1$. Recalling that $X = B^* X'$, the feature vector $x_1$ of the hyperedge $e_1$ is then calculated as:

$$x_1 = \vec{B}^*_{1u} \cdot x_u + \vec{B}^*_{1v} \cdot x_v + \vec{B}^*_{1c} \cdot x_c = 2 + i.$$

In the case where $\vec{H} = H$, i.e., when the hypergraph is undirected, we have $\vec{B}^* = B^\top$. The feature vector $x_1$ of the hyperedge $e_1$ is then calculated as:

$$x_1 = B_{1u} \cdot x_u + B_{1v} \cdot x_v + B_{1c} \cdot x_c = 3.$$

As illustrated by this example , in the specific case of a directed line graph, the feature vector can feature both real and imaginary components, depending on the topology of the hypergraph encoded by $\vec{B}$.

## F   CONFUSION MATRIX

We report the confusion matrices of `Dataset-1` in Figure 5 and `Dataset-2` in Figure 6. We can extract some insights from these two matrices, in particular:

- `Dataset-1.` DLGNet achieves a maximum performance of 88% in classifying the *Class 5* (Protections). However, its performance drops for *Class 4* and *Class 9* (Heterocycle formations and Functional group interconversions), where it correctly predicts only 44% and 41%, respectively.

- `Dataset-2.` DLGNet accurately classifies the sub-classes relative to the C–C bond formations (Reizman Suzuki Cross-Coupling and Imidazole-aryl coupling), as well as the Amide bond formations. On the other hand, the remaining three N-arylation sub-classes are poorly discriminated. This behavior can likely be attributed to the fact that the former are derived from different collections of Pd-catalyzed cross-coupling reactions, each exhibiting distinct features in terms of participant molecules (e.g. imidazole compounds). In contrast, all of the elements in the N-arylation classes share the same reaction mechanism (Buchwald-Hartwig amination); this poses a greater challenge, which results in decreased accuracy when predicting the correct class.

## G   FROM A DIRECTED HYPERGRAPH TO THE DIRECTED LINE GRAPH LAPLACIAN

To illustrate the construction of the directed line graph and the associated Directed Line Graph Laplacian, consider a directed hypergraph $\vec{H} = (V, E)$ where the vertex set is $V = \{a, b, c, d, e\}$ and the hyperedge set is $E = \{e_1, e_2, e_3\}$. The incidence relationships are defined as follows:

- $H(e_1) = \{b, c\}, T(e_1) = \{a\}$,
- $H(e_2) = \{a, b\}, T(e_2) = \{d\}$,
- $H(e_3) = \{e\}, T(e_3) = \{d\}$.

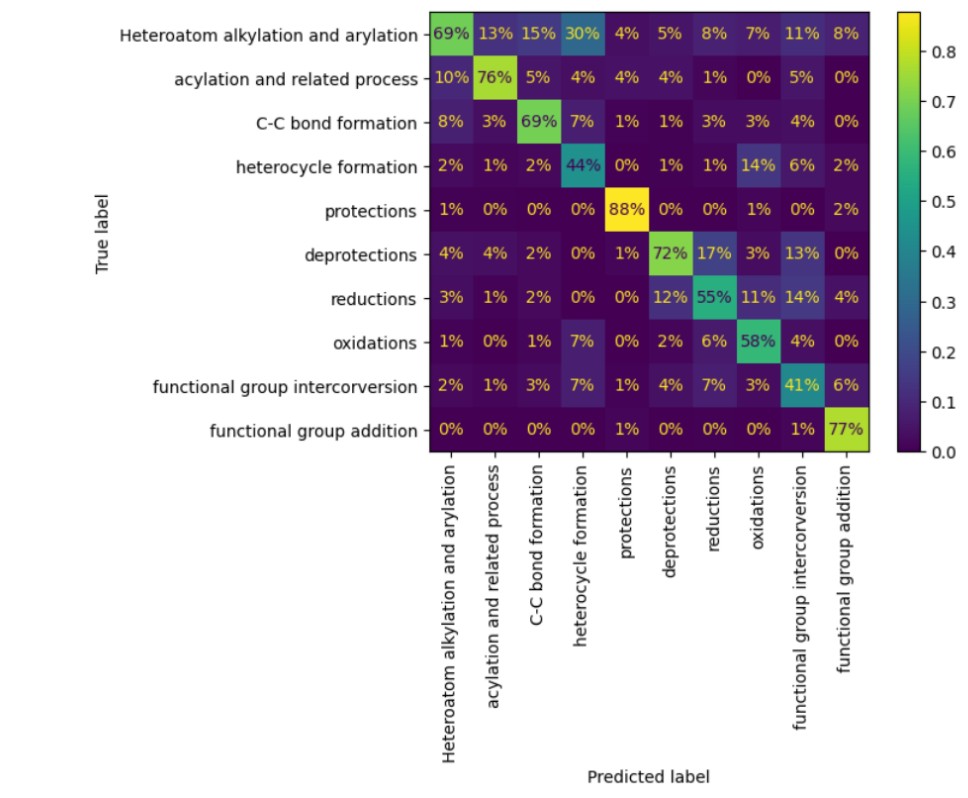

Figure 5: `Dataset-1` confusion matrix.

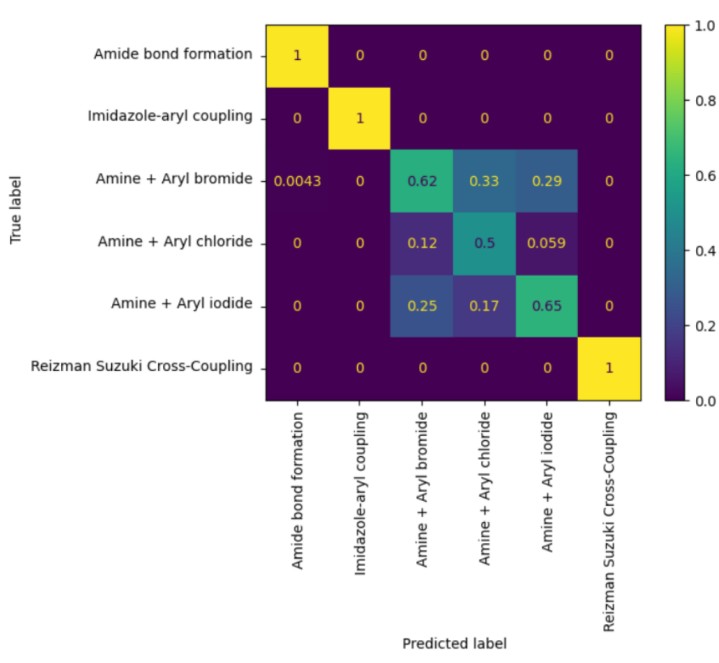

Figure 6: `Dataset-2` confusion matrix.

Each hyperedge is assigned a unit weight (i.e., $W = I$). The cardinalities (densities) of the hyperedges are $\delta_{e_1} = 3$, $\delta_{e_2} = 2$, and $\delta_{e_3} = 2$.

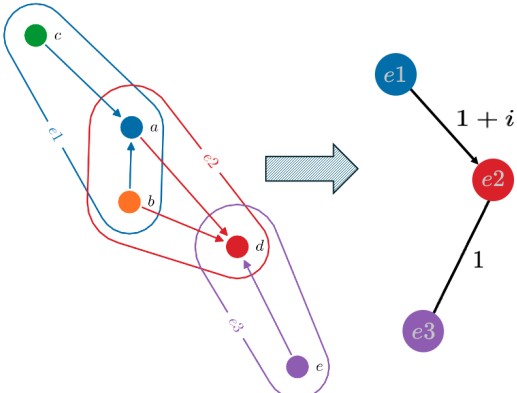

Figure 7: An example illustrating the transformation of a hypergraph (left) into its corresponding directed line graph (right).

We construct $DLG(\vec{H})$ using the following matrices: the incidence matrix $\vec{B}$, its conjugate transpose $\vec{B}^*$, the vertex degree matrix $D_v$, and the hyperedge degree matrix $D_e$. The incidence matrix $\vec{B}$ and its conjugate transpose are:

$$\vec{B} = \begin{bmatrix} -i & 1 & 0 \\ 1 & 1 & 0 \\ 1 & 0 & 0 \\ 0 & -i & -i \\ 0 & 0 & 1 \end{bmatrix} \quad \vec{B}^* = \begin{bmatrix} i & 1 & 1 & 0 & 0 \\ 1 & 1 & 0 & i & 0 \\ 0 & 0 & 0 & i & 1 \end{bmatrix}.$$

The vertex degree matrix $D_v$ and the hyperedge degree matrix $D_e$ are given by:

$$D_v = \begin{bmatrix} 2 & 0 & 0 & 0 & 0 \\ 0 & 2 & 0 & 0 & 0 \\ 0 & 0 & 1 & 0 & 0 \\ 0 & 0 & 0 & 2 & 0 \\ 0 & 0 & 0 & 0 & 1 \end{bmatrix} \quad D_e = \begin{bmatrix} 3 & 0 & 0 \\ 0 & 3 & 0 \\ 0 & 0 & 2 \end{bmatrix}.$$

Using these matrices, the adjacency matrix $A$ of the directed line graph $DLG(\vec{H})$ is:

$$A = \vec{B}^* \vec{B} - D_e = \begin{bmatrix} 0 & 1+i & 0 \\ 1-i & 0 & 1 \\ 0 & 1 & 0 \end{bmatrix}. \tag{14}$$

By Definition 1, the directed line graph $DLG(\vec{H})$ has three vertices, corresponding to the hyperedges $e_1$, $e_2$, and $e_3$ of the original hypergraph $\vec{H}$. An edge exists between two vertices in $DLG(\vec{H})$ if and only if their corresponding hyperedges in $\vec{H}$ are incident. In the specific example (illustrated in Figure 7), DLG($\vec{H}$) contains two edges, whose direction and weight are determined by the adjacency matrix $A$ (in equation 14. Without loss of generality, we consider the upper triangular part of $A$ to assign weights to the edges and define the directions: In the example considered, one edge will be directed and have a weight equal to $1+i$ (i.e. $e_1 \overset{1+i}{\to} e_2$), the other edge will be undirected and have a weight equal to $1$ ($e_2 \overset{1}{-} e_3$).

Using the equation 9. we can calculate the proposed Directed Line Graph Laplacian $\vec{\mathbb{L}}_N$ as follows:

$$\vec{\mathbb{L}}_N = I - \vec{\mathbb{Q}}_N := \vec{D}_e^{-\frac{1}{2}} \vec{B}^* \vec{D}_v^{-1} \vec{B} \vec{D}_e^{-\frac{1}{2}} = \begin{bmatrix} 0.333 & -0.167 - 167i & 0 \\ -0.167 + 0.167i & 0.5 & -0.204 \\ 0 & -0.204 & 0.25 \end{bmatrix}.$$

By inspecting $\vec{\mathbb{L}}_{\mathbb{N}}$, one can observe that it encodes the elements of the hypergraph $\vec{H}$ in the following way:

1. The real components of off-diagonal entries in $\vec{\mathbb{L}}_{\mathbb{N}}$ encode the fact that, in the underlying hypergraph $\vec{H}$, the vertex belongs to the head set or tail set simultaneously in two different hyperedges. For example, $\vec{\mathbb{L}}_{\mathbb{N}}(2,3) = -0.204$ indicates that $H(e_2) \cap H(e_3) \neq \emptyset$ or $T(e_2) \cap T(e_3) \neq \emptyset$. In this specific case, $T(e_2) \cap T(e_3) = \{d\}$. Similarly, $\Re(\vec{\mathbb{L}}_{\mathbb{N}}(1,2)) = -0.167$ arises from the fact that $e_1$ and $e_2$ share the vertex $b$ in their head sets.

2. The imaginary component captures the hyperedge directionality based on the underlying hypergraph $\vec{H}$, where a node belongs to the head set of one hyperedge and the tail set of another. For example, $\Im(\vec{\mathbb{L}}_N(1,2)) = -\Im(\vec{\mathbb{L}}_N(2,1)) = -0.167$, indicating that $a \in T(e_1) \cap H(e_2)$.

3. The absence of any relationships between hyperedges $e1$ and $e3$ is encoded by 0 in $DGL(\vec{H})$. Specifically, $\vec{\mathbb{L}}_N(1,3) = \vec{\mathbb{L}}_N(3,1) = 0$.

4. The *self-loop information* (a measure of how strongly the feature of a vertex depends on its current value within the convolution operator) is encoded by the diagonal of $\vec{\mathbb{L}}_N$.

## H  OUR INCIDENT MATRIX $\vec{B}$

Utilizing our incidence matrix $\vec{B}$, where $\vec{B}_{ve} = -i$ if $v \in T(e)$, with complex numbers allows us to encode directionality and construct a Laplacian, the Directed Line Graph Laplacian, that is Hermitian and meets the necessary properties for applying a spectral-based approach.

If, instead, we had chosen to use $B_{ve} = -1$ if $v \in T(e)$, we would have lost the directionality of the hypergraph. To illustrate, consider (for simplicity—this can be observed also for more hypergraphs) a graph with nodes $1, 2$ and edges $e_1 = (2, 1)$ and $e_2 = (3, 2)$.

We have $\vec{B} = \begin{pmatrix} -i & 0 \\ 1 & -i \\ 0 & 1 \end{pmatrix}$ and $B = \begin{pmatrix} -1 & 0 \\ 1 & -1 \\ 0 & 1 \end{pmatrix}$.

The Laplacian matrix we use in DLGNet is $\vec{L} = \begin{pmatrix} 2 & -i \\ i & 2 \end{pmatrix}$. The Laplacian matrix using the $B$ the reviewer suggests reads $L = \begin{pmatrix} 2 & -1 \\ 1 & 2 \end{pmatrix}$.

Let us recall that the nodes of both matrices correspond to the edges of the graph. Therefore, $\vec{L}_{12}$ indicates the presence of the directed line graph edge $(e_1, e_2)$, which captures the topology of the original graph where edge $e_2$ is seen before edge $e_1$ in a path from node 3 to node 1.

Differently, since $L_{12} = L_{21}$, in $L$ such a directional information is completely lost.

Regarding the solution proposed in (Ma et al., 2024), this approach defines two separate incidence matrices: one for tail elements $B_T = \begin{pmatrix} 1 & 0 \\ 0 & 1 \\ 0 & 0 \end{pmatrix}$ and one for head elements and $B_H = \begin{pmatrix} 0 & 0 \\ 1 & 0 \\ 0 & 1 \end{pmatrix}$.

The Laplacian matrix using $B_T$ and $B_H$ reads $L = \begin{pmatrix} 0 & 0 \\ 1 & 0 \end{pmatrix}$.

This matrix $L$ is not symmetric, which does not fit the framework in which we operate in our paper, in which we are designing a spectral-based convolutional operator. This type of GNN does not permit the use of a non-symmetric matrix, as it requires an eigenvalue decomposition of the Laplacian matrix with real eigenvalues. Thanks to the adoption of a complex-valued $\vec{B}$, our proposed Laplacian matrix is Hermitian and, therefore, admits the required eigenvalue decomposition.