# OpenReview forum: "DLGNet: Hyperedge Classification through Directed Line Graphs for Chemical Reactions"
_ICLR.cc/2025/Conference — Submitted to ICLR 2025_

### Official Review · Reviewer_YKY8 · 2024-10-29

**Soundness:** 3
**Presentation:** 3
**Contribution:** 3
**Rating:** 8
**Confidence:** 4

**Summary:**

The paper introduced a novel method that convert the directed hypergraph into Directed Line Graph (DLG), where each hyperedge is transformed into a vertex in the DLG. The DLG Laplacian is proved to be positive semidefinite and spectral-based GNN is introduced for the DLG. This novel representation and associated GNN operation are applied for chemical reaction classification problem.

**Strengths:**

1. The study is comprehensive. Starting from the mathematical definition of the directed hypergraph, the laplacian is derivated and related properties are proved. Then the convolution operation is introduced and complexity is also analyzed.

2. The paper includes many details including the public-available code, proofs of theorems, complexity analysis, dataset and model setting in the Appendix, which is very helpful to understand the paper.

3. The importance of directionality are well assessed in ablation study, which supports the motivation of the proposed method.

**Weaknesses:**

1. The baseline performance for reaction classification is quetionable:
   The F1 scores reported in Table 1 for baseline methods are all at random guess level (e.g. F1<0.1 for 10-class classification). However in other reaction classification datasets and baseline methods, the F1 could be larger than 0.8 even for 1000 classes. [1]
   Please verify the baseline metrics, and compare with the baseline methods which could get better prediction performance.

2. The definition of node and hyperedge for chemical reaction is not clear:
   In Section 1 Line 72, author mentions tackle the reaction classification problem as hyperedge classification task. The Figire 1 also indicates each molecule is a node and each reaction is a hyperedge.
   However in Figure 2 Dataset-1 and Dataset-2, the head and tail are for single molecule. In Section 2 Line 130, the head set and tail set are for hyperedge e, which means the hyperedge is a single molecule.
   The author should make the definition consistent.

3. Proofs could be more clear:
   In line 740 and 743, the second term is transformed from sum(i,j \in E) to sum(i,j \in E: i<=j) by mapping the lower triangle elements to upper triangles. However, the diagonal terms are introduced twice and should be subtracted from the new format.
   In line 743 and line 745, how the first term was transformed should be further clarified.

References:
[1] Li, Anchen, Elena Casiraghi, and Juho Rousu. "Chemical reaction enhanced graph learning for molecule representation." Bioinformatics 40.10 (2024): btae558.

**Questions:**

1. High Level:
   After converting the directed hypergraph to DLG, the relationship of vertices belongs to the same hyperedge are not reflected in DLG anymore and the representation of the original graph is not updated during learning.
   In Line 294: the feature matrix for the vertices of DLG is a simple aggregation of original feature matrix.
   Will this exacerbate the learning ability?

2. The Dataset description could be enriched:
	In Figure 1 Dataset-1 example, only 2 reactnats are involved and it seems only one hyperedge for reaction remains. Then the graph convolution layer becomes linear layer since no neighbors are included.
	Could you provide the statistics of the reactions in the datasets (e.g. the average number of vertices and average node degree in DLG)?

3. Experiments:
   Line 408-409, with 5-fold cross-validation, why do we still need 50%/25%/25% splition for train/val/test?

Typo:
1. Line 308, the output shape should be m x 2c', instead of n x 2c'.
2. Line 742, the * for first B(u,j) should be removed.

---

> ### Author Response · Authors · 2024-11-20
> **Response to Reviewer 4 (parts 1/2)**
>
> **The baseline performance for reaction classification is quetionable: The F1 scores reported in Table 1 for baseline methods are all at random guess level (e.g. F1<0.1 for 10-class classification). However in other reaction classification datasets and baseline methods, the F1 could be larger than 0.8 even for 1000 classes. [1] Please verify the baseline metrics, and compare with the baseline methods which could get better prediction performance.**
>
> As mentioned in Section 5.2, we face a class imbalance problem in our experiments due to the fact that the classes do not occur with equal numerosity in the dataset. In line with common practices in machine learning, we use the F1-score---the harmonic mean of precision and recall---because it more accurately reflects the model's performance in imbalanced data situations than accuracy would.\
> We kindly point out that: i) [1] was not available for comparison at the time of writing this paper, since it was published in October 2024 after our ICLR submission; and ii) [1] does not use the F1-score (see Table 2 in [1]), opting for accuracy and recall instead. Without additional metrics like precision or additional information on class imbalance, a direct comparison between their results and ours is challenging, especially given that different datasets were used in both cases.
>
>  [1] Chemical reaction enhanced graph learning for molecule representation. Bioinformatics 40.10 (2024): btae558
>
> **The definition of node and hyperedge for chemical reaction is not clear: In Section 1 Line 72, author mentions tackle the reaction classification problem as hyperedge classification task. The Figure 1 also indicates each molecule is a node and each reaction is a hyperedge. However, in Figure 2 Dataset-1 and Dataset-2, the head and tail are for single molecule. In Section 2 Line 130, the head set and tail set are for hyperedge e, which means the hyperedge is a single molecule. The author should make the definition consistent.**
>
> As described in Section 1, we define each molecule as a node and its interaction with other molecules as a directed hyperedge. To better clarify this point, we have slightly changed Figure 1 (and its caption) and added a new reference to the description on how we model the nodes and hyperedges for chemical reactions in Section 5.3.
>
> In the example of Figure 2, there are multiple reactants for Dataset-1 and Dataset-2, while the product is only one. The hyperedge consists of all the reactants and products. Differently, in Dataset-3 we have two products.
>
> To better clarify the definition in line 130, we have rewritten the sentence:\
> **Original.** *We define a directed hypergraph $\vec H$ as a hypergraph where each hyperedge $e \in E$ is partitioned in a head set $H(e)$ and a tail set $T(e)$*.\
> **Revised.** *we define a directed hypergraph $\vec H$ as a hypergraph where each node in each hyperedge $e \in E$ belongs to either a head set $H(e)$ or a tail set $T(e)$*.
>
> **Proofs could be more clear: In line 740 and 743, the second term is transformed from $\sum(i,j \in E)$ to $\sum(i,j \in E: i<=j)$ by mapping the lower triangle elements to upper triangles. However, the diagonal terms are introduced twice and should be subtracted from the new format. In line 743 and line 745, how the first term was transformed should be further clarified.**
>
> We thank the reviewer for this comment. We have completely revised the proof in Appendix B, providing a detailed clarification of the steps in the proof.

---

> > ### Comment · Reviewer_YKY8 · 2024-11-25
> >
> > Dear authors,
> >
> > Thank you for your responses. The response answered some of my questions.
> > I still have following concerns.
> >
> > 1. Baseline score:
> >     As you mentioned, F1 is the harmonic mean of precision and recall so we can do simple calculation and get the F1 score for the table 2 in [1], and it is still > 0.8. Although the referenced paper[1] is released recently, the baseline method included in the paper [2][3] are released couple years ago and the baseline F1 scores calculated from precision and recall are always >0.8.
> >     For multi-classification task, there is no uniformed definition of F1 score. In scikit-learn[4], there are 5 different ways to calculate the F1-score.  F1< 0.1 for 10-class classification task does not makes sense to me, no matter which definition it is. Please share the detailed definition that you used for F1-score, and indicate why imbalanced dataset could make the metric < 0.1. The detailed information, including confusion matrix and precision/recall/F1/sample size for each class, and the derivation of final F1 metric could help.
> >     (Here 0.8 or 0.1 are based on precision and recall are both percentage and the scales are between 0-1).
> >
> > 2. Definition of head/tail set
> >     According to the updated definition "we define a directed hypergraph $\vec H$ as a hypergraph where each node in each hyperedge $e \in E$ belongs to either a head set $H(e)$ or a tail set $T(e)$." The head/tail set is for edge, I am not sure how node can belong to an edge set. Please clarify.
> >
> > 3. Pooling:
> >     I understand the pooling operation for graph node representation. This usually happens after convolution operation on original node representation, such as in [5]. The topology structure are preserved and the model could update the original node representation first then aggregate to higher level (e.g node-group level or graph level).
> >     In the proposed network, the convolution happens after the aggregation, which makes the original node representation is not updated through learning, and original topology is totally discarded after the fixed aggregation. This is OK and it could be fixed by adding GCN layer before aggregation to make the model more flexible.
> >     This question is also related to the first question about baseline score. The F1 score of 0.6 is OK, but I am not sure if it could surpass the baseline methods.
> >
> >
> > [1] Chemical reaction enhanced graph learning for molecule representation. Bioinformatics 40.10 (2024): btae558
> > [2] Jaeger S, Fulle S, Turk S. Mol2vec: unsupervised machine learning ap- proach with chemical intuition. J Chem Inf Model 2018;58:27–35.
> > [3] Fabian B, Edlich T, Gaspar H et al. Molecular representation learning with language models and domain-relevant auxiliary tasks. arXiv, arXiv:2011.13230, 2020, preprint: not peer reviewed.
> > [4] https://scikit-learn.org/1.5/modules/generated/sklearn.metrics.f1_score.html
> > [5] Ying, Zhitao, et al. "Hierarchical graph representation learning with differentiable pooling." Advances in neural information processing systems 31 (2018).

---

> > > ### Author Response · Authors · 2024-11-26
> > > **Response to Reviewer 4  (parts 1/2)**
> > >
> > > **Baseline score: As you mentioned, F1 is the harmonic mean of precision and recall so we can do simple calculation and get the F1 score for the table 2 in [1], and it is still > 0.8. Although the referenced paper[1] is released recently, the baseline method included in the paper [2][3] are released couple years ago and the baseline F1 scores calculated from precision and recall are always >0.8. For multi-classification task, there is no uniformed definition of F1 score. In scikit-learn[4], there are 5 different ways to calculate the F1-score. F1< 0.1 for 10-class classification task does not makes sense to me, no matter which definition it is. Please share the detailed definition that you used for F1-score, and indicate why imbalanced dataset could make the metric < 0.1. The detailed information, including confusion matrix and precision/recall/F1/sample size for each class, and the derivation of final F1 metric could help. (Here 0.8 or 0.1 are based on precision and recall are both percentage and the scales are between 0-1).**
> > >
> > > We respectfully invite the reviewer to revisit Table 2 of paper [1], where the authors report only *Accuracy* and *Recall*. Since additional details, including Precision, are not provided, it is not possible to compute the F1 score. By definition, the F1 score is given by the following equation $F1 = 2 \times \frac{\text{Precision} \times \text{Recall}}{\text{Precision} + \text{Recall}}$ or using True Positives (TP), False Negatives (FN), and False Positives (FP).
> > >
> > > Regarding the models [2, 3], they are designed to learn vector representations of molecular substructures. While their evaluation in prior work focused on certain downstream tasks, we emphasize that [1] is the first to apply these models to the reaction classification task.
> > >
> > > In this study, we calculate the F1 score using the *macro average*, which is defined on the Scikit-learn documentation [4] as *calculating metrics for each label and finding their unweighted mean.*
> > >
> > > To demonstrate the correctness of our metric, we report the results on *Dataset-2* using the **HyperND** model. Below, we present the results related to the confusion matrix, F1 score, precision and recall.
> > >
> > > **Confusion Matrix**
> > >
> > > $$\text{Confusion Matrix} = \begin{bmatrix}3777 & 0 & 0 & 0 & 0 & 0 & 0 & 0 & 0 & 0 \\\\2976 & 0 & 0 & 0 & 0 & 0 & 0 & 0 & 0 & 0 \\\\1356 & 0 & 0 & 0 & 0 & 0 & 0 & 0 & 0 & 0 \\\\234 & 0 & 0 & 0 & 0 & 0 & 0 & 0 & 0 & 0 \\\\173 & 0 & 0 & 0 & 0 & 0 & 0 & 0 & 0 & 0 \\\\2113 & 0 & 0 & 0 & 0 & 0 & 0 & 0 & 0 & 0 \\\\1185 & 0 & 0 & 0 & 0 & 0 & 0 & 0 & 0 & 0 \\\\193 & 0 & 0 & 0 & 0 & 0 & 0 & 0 & 0 & 0 \\\\458 & 0 & 0 & 0 & 0 & 0 & 0 & 0 & 0 & 0 \\\\39 & 0 & 0 & 0 & 0 & 0 & 0 & 0 & 0 & 0\end{bmatrix}$$
> > >
> > > In this case, the model predicts all instances as belonging to the largest class.
> > >
> > > **F1 Score**
> > >
> > > $\text{F1-score} = \begin{bmatrix}0.46 & 0.0 & 0.0 & 0.0 & 0.0 & 0.0 & 0.0 & 0.0 & 0.0 & 0.0\end{bmatrix}$. The **macro F1 score** is 0.046, which is the average F1 score calculated across all classes.
> > >
> > > **Precision**
> > >
> > > $\text{Precision} = \begin{bmatrix}0.30, 0.0, 0.0, 0.0, 0.0, 0.0, 0.0, 0.0, 0.0, 0.0\end{bmatrix}$. The **macro precision** is 0.030.
> > >
> > > **Recall**
> > >
> > > $\text{Recall} = \begin{bmatrix}1.0, 0.0, 0.0, 0.0, 0.0, 0.0, 0.0, 0.0, 0.0, 0.0\end{bmatrix}$. The **macro recall** is 0.1.
> > >
> > > As demonstrated in the example, when the model predicts all outputs as the first class (see confusion matrix), the obtained macro F1 score is less than 0.1. If the reviewer considers it appropriate, we can report the weighted F1 score for each model in the Appendix.
> > >
> > >
> > >
> > > [1] Chemical reaction enhanced graph learning for molecule representation. Bioinformatics 40.10 (2024): btae558
> > >
> > > [2] Jaeger S, Fulle S, Turk S. Mol2vec: unsupervised machine learning ap- proach with chemical intuition. J Chem Inf Model 2018;58:27–35.
> > >
> > > [3] Fabian B, Edlich T, Gaspar H et al. Molecular representation learning with language models and domain-relevant auxiliary tasks. arXiv, arXiv:2011.13230, 2020, preprint: not peer reviewed.
> > >
> > > [4] https://scikit-learn.org/1.5/modules/generated/sklearn.metrics.f1\_score.html

---

> ### Author Response · Authors · 2024-11-20
>
> **High Level: After converting the directed hypergraph to DLG, the relationship of vertices belongs to the same hyperedge are not reflected in DLG anymore and the representation of the original graph is not updated during learning. In Line 294: the feature matrix for the vertices of DLG is a simple aggregation of original feature matrix. Will this exacerbate the learning ability?**
>
> The feature matrix of the vertices of the DLG (which is obtained through the topology-aware aggregation $X^{'} = \vec B^* X$, which is similar to a \textit{graph pooling operation}) is updated (the features are spectrally convoluted) through the weights ($\Theta_0$ and $\Theta_1$ in Equation (12)) that are learned during the model training process. The reviewer is correct in noting that neither the topology of the directed hypergraph nor that of the directed line graph changes during the training process. This is quite standard in the graph-convolutional network literature.  Thanks to the fact that the feature aggregation in our model is performed using the $\vec{B}^{*}$ matrix, such an aggregation preserves the head-tail directional information of the nodes in the hyperedge using complex numbers, allowing the model to adequately learn the relationships between the features.
>
> **The Dataset description could be enriched: In Figure 1 Dataset-1 example, only 2 reactants are involved and it seems only one hyperedge for reaction remains. Then the graph convolution layer becomes linear layer since no neighbors are included. Could you provide the statistics of the reactions in the datasets (e.g. the average number of vertices and average node degree in DLG)?**
>
> The figure reports just a simple example of the possible interactions within the datasets for illustration purposes. This does not mean that there are no neighborhoods; indeed, one of the reactants or the product may belong to other hyperedges.
> We collected the following statistics:
>
> **Table:** Statistics on Hypergraphs and Directed Line Graphs
>
> |           | Hypergraph |       | Directed Line Graph |               |
> | --------- | :--------: | :---: | ------------------- | ------------- |
> |           |     n      | \|E\| | Density             | Vertex Degree |
> | Dataset-1 |   100523   | 50016 | 0.10%               | 48.05         |
> | Dataset-2 |    956     | 3021  | 43.29%              | 1036          |
> | Dataset-3 |    670     |  649  | 73%                 | 511.20        |
>
> As illustrated in the table above, it is highly unlikely that the graph convolutional layer reduces to a linear layer due to the absence of neighbors. Instead, it is very likely for each node to have neighboring connections.
>
> **Experiments: Line 408-409, with 5-fold cross-validation, why do we still need 50\%/25\%/25\% spliting for train/val/test?**
>
> Actually, we generate 5 random data splits, using a 50%/25%/25% split for training, validation, and testing. We have better clarified this in the paper.

---

> ### Author Response · Authors · 2024-11-26
> **Response to Reviewer 4 (parts 2/2)**
>
> **Definition of head/tail set According to the updated definition "we define a directed hypergraph $\vec H$ as a hypergraph where each node in each hyperedge belongs to either a head set $H(e)$ or a tail set $T(e)$." The head/tail set is for edge, I am not sure how node can belong to an edge set. Please clarify.**
>
> As defined in Section 2, a hypergraph $H = (V, E)$ is a generalization of a graph in which an edge can connect any number of vertices. Specifically, $V$ represents the set of vertices (or nodes), and $E \subseteq 2^V \setminus \\{\\}$ is the (nonempty) set of hyperedges. This means that each hyperedge $e$ is defined as a set of nodes (only those involved in the chemical reaction the hyperedge represents). In contrast, in a graph (also called a 2-uniform hypergraph), an edge (i.e., the hyperedge) connects exactly two vertices.
> In the case of a directed hypergraph, each hyperedge is divided into two distinct subsets, $T(e)$ and $H(e)$, which represent the products and reactants, respectively, and define the direction of the hyperedge.
>
> For example, consider a hypergraph $H = (V, E)$ where the vertex set is $V=\\{a, b, c, d, e\\}$ and the hyperedge set is $E = \\{e_1, e_2, e_3\\}$. The hyperedges are defined as follows:
> - $e_1 = \\{b, c, a\\}$,
> - $e_2 = \\{a, b, d\\}$,
> - $e_3 = \\{e, d\\}$.
>
> Now consider a directed hypergraph $\vec H = (V, E)$ with the same vertex set $V = \\{a, b, c, d, e\\}$ and hyperedge set $E = \\{e_1, e_2, e_3\\}$. The hyperedge relationships are defined by splitting the nodes of each hyperedge into $T(e)$ (tail) and $H(e)$ (head) as follows:
>  - $H(e_1) = \\{b, c\\}$; $T(e_1) = \\{a\\}$,
>  - $H(e_2) = \\{a, b\\}$; $T(e_2) = \\{d\\}$,
>  - $H(e_3) = \\{e\\}$; $T(e_3) = \\{d\\}$.
>
>
> **Pooling: I understand the pooling operation for graph node representation. This usually happens after convolution operation on original node representation, such as in [5]. The topology structure are preserved and the model could update the original node representation first then aggregate to higher level (e.g node-group level or graph level). In the proposed network, the convolution happens after the aggregation, which makes the original node representation is not updated through learning, and original topology is totally discarded after the fixed aggregation. This is OK and it could be fixed by adding GCN layer before aggregation to make the model more flexible. This question is also related to the first question about baseline score. The F1 score of 0.6 is OK, but I am not sure if it could surpass the baseline methods.**
>
> Regarding the baselines, as detailed in the supplementary materials and Section 5.3, the pooling operation is applied after convolutions on the hypergraph. Therefore, the original topology is not discarded. In contrast, our method applies convolution after transforming the features. However, this does not pose an issue because the topology on which our method operates is different from the original directed hypergraph. Specifically, we perform convolutions on the directed line graph. While applying convolution directly on the directed hypergraph could be explored, the layers in DLGNet are designed to update the features effectively, as demonstrated by the results in the experimental section.
>
> We also do not fully understand the reviewer's statement, *"I am not sure if it could surpass the baseline methods,"* as our method consistently outperforms all the baseline methods. If the reviewer is specifically referring to baselines [2, 3], a direct comparison is not feasible due to differences in experimental setups: the datasets used in [2, 3] are different, and their reported metrics include only *Accuracy* and *Recall*, without providing the F1 score.
>
> [1] Chemical reaction enhanced graph learning for molecule representation. Bioinformatics 40.10 (2024): btae558
>
> [2] Jaeger S, Fulle S, Turk S. Mol2vec: unsupervised machine learning ap- proach with chemical intuition. J Chem Inf Model 2018;58:27–35.
>
> [3] Fabian B, Edlich T, Gaspar H et al. Molecular representation learning with language models and domain-relevant auxiliary tasks. arXiv, arXiv:2011.13230, 2020, preprint: not peer reviewed.
>
> [4] https://scikit-learn.org/1.5/modules/generated/sklearn.metrics.f1\_score.html
>
> [5] Ying, Zhitao, et al. "Hierarchical graph representation learning with differentiable pooling." Advances in neural information processing systems 31 (2018).

---

> > ### Comment · Reviewer_YKY8 · 2024-11-27
> >
> > Thank you for the response.
> > The updated information and example makes the paper more clear and resolved my concerns. Have raised the score accordingly.

---

### Official Review · Reviewer_18Zj · 2024-11-03

**Soundness:** 3
**Presentation:** 3
**Contribution:** 3
**Rating:** 6
**Confidence:** 3

**Summary:**

In the field of organic synthesis, accurately predicting reaction types can help chemists design and optimize synthetic routes. This article proposed DLGNet. This is a novel approach for classifying chemical reactions by leveraging a new spectral-based graph neural network for hypergraphs. DLGNet utilizes a directed line graph Laplacian operator Hermitian matrix to encode the directionality and connectivity between hyperedges. The Hermitian property and positive semidefiniteness of this matrix make it suitable for spectral convolution, which the DLGNet leverages to enhance classification performance.

**Strengths:**

1. The chemical reactions often involve interactions between multiple substances, rather than simple binary relationships (e.g., reactants-products). Using hypergraphs to represent and analyze chemical reactions can more naturally capture this multivariate relationship.

2. The mathematical framework is innovative and critical. The introduction of "Directed Line Graph Laplacian" is a key mathematical contribution. By constructing a Hermitian matrix, complex-values Laplacian, the authors enable the network to capture both the directionality and connectivity of hyperedges, a sophisticated approaches for chemical reaction classifications.

3. DLGNet has been conducted extensive experiments on three different and diverse real-world datasets. Also including a robust ablation study to demonstrate the importance of directionality in the model. These all show the benefits of the proposed methods.

**Weaknesses:**

1. The use of Hermitian Laplacian matrix is mathematically convenient. However, using non-Hermitian matrix on the directed graph neural networks may sometimes provide more flexibility in encoding the directionality.

2. The model currently only relies on the molecular Morgan Fingerprints for the node features. However, consider more molecular features, such as electronic descriptors and three dimensional conformations, would provide the model a more comprehensive understanding of chemical reactions.

3. It would be beneficial to add a figure of the DLGNet model architecture in the main text. This would  help readers better understand the model.

**Questions:**

1.  Add a figure of the DLGNet model architecture in the main text.

2. Although the authors compared the model performance of DLGNet to other published graph neural networks. It would be beneficial to illustrate the differences between DLGNet and the other models to provide readers a clearer understanding.

---

> ### Author Response · Authors · 2024-11-20
> **Response to Reviewer 3**
>
> We thank the reviewer for reviewing our paper and providing their comments.
>
> **The use of Hermitian Laplacian matrix is mathematically convenient. However, using non-Hermitian matrix on the directed graph neural networks may sometimes provide more flexibility in encoding the directionality.**
>
> The existence of an eigenvalue decomposition yielding real eigenvalues is essential for applying the Fourier transform and the entire derivation proposed in [1], which is crucial for developing a principled spectral graph convolutional operator.
> Complex-valued Hermitian matrices admit an eigenvalue decomposition with real eigenvalues, and, if carefully designed, they can encode the direction of the relationship between nodes in the sign of the imaginary part.
> Such matrices already offer enhanced flexibility in comparison to their Hermitian counterparts featuring only real elements, with a positive effect on the overall performance of the network.
>
> While alternative approaches might, at least in theory, provide the network with greater flexibility (and thus greater scope for method design), the results of this study, as well as those of [2, 3], demonstrate that spectral convolutional operators based on Laplacians constructed from complex-valued Hermitian matrices outperform alternative approaches, including random walk-based and spatial-based models, in the tasks under consideration.
>
> Such a discussion, albeit interesting, appears to extend beyond the scope of the present paper, not least due to space limitations. However, we remain open to considering the addition of any other spectral method using non-Hermitian matrices for the definition of the Laplacian that the reviewer may wish to suggest.
>
> [1] Semi-supervised classification with graph convolutional networks. ICLR 2017.\
> [2] MagNet: A Neural Network for Directed Graphs. NeurIPS 2021.\
> [3] Sigmanet: One laplacian to rule them all. AAAI 2023.
>
>
> **The model currently only relies on the molecular Morgan Fingerprints for the node features. However, consider more molecular features, such as electronic descriptors and three dimensional conformations, would provide the model a more comprehensive understanding of chemical reactions.**
>
> We thank the reviewer for this suggestion. This suggestion is indeed in line with our current intention to enrich the features we use to describe the molecules representing the nodes in our hypergraph. We will certainly consider this for our future work.
>
> **It would be beneficial to add a figure of the DLGNet model architecture in the main text. This would help readers better understand the model.**
>
> We have added a new figure (Figure 2 in the main paper) illustrating the architecture of DGLNet.
>
> **Although the authors compared the model performance of DLGNet to other published graph neural networks. It would be beneficial to illustrate the differences between DLGNet and the other models to provide readers a clearer understanding.**
>
> We have added Section 5.2 where we better illustrate the differences between our model and the others. We have also added a numerical illustration in Appendix H, demonstrating the difference between our incident matrix $\vec{B}$, and other proposals.

---

> > ### Comment · Reviewer_18Zj · 2024-11-22
> > **Feedbacks to the authors rebuttal letter**
> >
> > Dear authors,
> >
> > Thank you for your responses. For the Hermitian Laplacian matrix, model architecture, baseline models discussions are more clear than the original submission.
> >
> > Even though DLGNet performs better than other hypergraph and graph neural networks. It is still not very clear about how the Hermitian Laplacian matrix handle the model's flexibility. Also, the inputs are just the molecular Morgan Fingerprints, which can not fully represent all the information of molecules. DLGNet model performance still need to be further verified by consider more molecular information, such as atom type, charge, chirality, aromatics, and more.
> >
> > Therefore, I think DLGNet still has some distance to go to reach point 8 (good paper).
> >
> > Best regards.

---

> > > ### Author Response · Authors · 2024-11-25
> > >
> > > We thank the reviewer for appreciating our responses.
> > >
> > > Regarding the flexibility, we remark that the use of the Hermitian matrix mathematically satisfies the properties required by [1] to apply graph convolutions, while, on a practical level, it enables the definition of a behavior that captures the direction of the interactions within the hypergraph. Via the imaginary part, which is skew-symmetric (i.e., $\Im(A)^\top = -\Im(A)$), we can effectively encode directionality. For instance, in the graph in Figure 7 in the Appendix G, the matrix $A$ of $DLG(\vec H)$, $A = \vec{B}^* \vec{B} - D_e = \begin{bmatrix}0 & 1+i  & 0 \\\\ 1-i & 0  & 1 \\\\ 0 & 1 & 0\end{bmatrix},$ has an nonzero imaginary component in positions $(1,2)$ and $(2,1)$ of opposite signs, indicating a direction from edge $e_1$ to edge $e_2$. Due to the fact that the imaginary component allows us to encode directionality, a positive message passing takes place form $e_1 \rightarrow e_2$ and a negative one takes place in the opposite direction.
> > >
> > > The goal of our paper is to explore the potential of our proposed mathematical framework, highlighting its strengths (e.g., convolutions directly on the directed line graph, handling directionality) compared to other architectures under identical initial conditions, i.e., with the same features encoding the characteristic of the molecule. We are aware that other features exist to characterize molecules and their properties. However, exploring such features was not the main goal of this specific work. As future direction we aim to pursue is to investigate different initial configurations to identify the optimal combination of features for improving model performance and better describing molecular properties. We believe that the addition of extra features could only lead to an even better performance than the one we obtained with the features we used.
> > >
> > > [1] Semi-supervised classification with graph convolutional networks. ICLR 2017.

---

> > > > ### Comment · Reviewer_18Zj · 2024-12-03
> > > >
> > > > Dear authors,
> > > >
> > > > Thank you for your efforts and explanations. I checked through the article again, and would still keep my original score.
> > > >
> > > > Best regards.

---

### Official Review · Reviewer_nufw · 2024-11-04

**Soundness:** 3
**Presentation:** 1
**Contribution:** 2
**Rating:** 5
**Confidence:** 4

**Summary:**

This paper addresses the chemical reaction classification problem by introducing the Directed Line Graph (DLG) associated with a given directed hypergraph. It also deduces a Directed Line Graph Network (DLGNet) which achieves impressive performance on the chemical reaction classification task.

**Strengths:**

- This paper proposed a new direct hypergraph neural network for chemical reaction prediction.  This application is interesting and the proposed method sounds technique.

**Weaknesses:**

- The paper says it only focuses on modeling reaction structures without considering any form of hypergraph learning methods. However, it is unclear whether some traditional methods have been proposed for this task. If there exists, it should be added to support the efficiency of your method. Further, the proposed method is obviously can be used for link prediction, can you explain why it cannot achieve good performance on the task?
- Lack of details of how to model the reaction structures to hypergraph edge prediction task. Is it the node of the hypergraph is the molecular? This problem definition should be clearly shown in the main paper. Further, why DLG design can specifically work for this reaction task? In our understanding, the specific application paper should consider some domain knowledge (such as chemical reaction prior information) to make the proposed approach convincing.
- The design of the $Bve=-i, if v\in T(e)$ is weird, what is the theoretical or intuitive motivation? What if we use $Bve=-1, if v\in T(e)$ or we still use the B as described in the typical method like [1].
- This paper lacks some necessary baselines and is without discussion with some related works. For example [1,2,3]. Besides, the Magnetic Laplacian also be used in [4].

minor: Some key matrices should describe their dimensions for ease of reading.

[1]Directed Hypergraph Representation Learning for Link Prediction, AISTATS 2024
[2] Hypergraph Convolutional Networks via Equivalency between Hypergraphs and Undirected Graphs, ICML workshop 2022
[3] Directed hypergraph neural network, 2020
[4] Unified Random Walk, Its Induced Laplacians and Spectral Convolutions for Deep Hypergraph Learning, 2022

**Questions:**

see weakness

---

> ### Author Response · Authors · 2024-11-20
> **Response to Reviewer 2 (parts 1/3)**
>
> We thank the reviewer for reviewing our paper and providing their comments.
>
> **The paper says it only focuses on modeling reaction structures without considering any form of hypergraph learning methods. However, it is unclear whether some traditional methods have been proposed for this task. If there exists, it should be added to support the efficiency of your method.**
>
> We were not aware of any prior work specifically addressing the chemical reaction classification task. In our paper we do consider hypergraph learning methods; indeed, our proposed DLGNet is one such method: it takes a hypergraph as input which encodes molecules in its nodes and chemical reactions in its hyperedges and produces as output the chemical-reaction class of each chemical reaction (hyperedge). In doing so, it realizes a graph convolution on the directed line graph associated with the input directed hypergraph, whose Laplacian matrix is constructed following our proposed Equation (9) (see also Equation (10) for the description of such a Laplacian matrix in scalar form).
>
> **Further, the proposed method is obviously can be used for link prediction, can you explain why it cannot achieve good performance on the task?**
>
> We fear there is a misunderstanding here. Link prediction and hyperedge classification are two fundamentally different problems: link prediction aiming to determine future connections between pairs of unconnected links based on existing connections---see [1] for reference. In contrast, as defined in Section 1, in this paper, we address the hyperedge classification problem, which involves predicting the class of the hyperedges. Specifically, in our case, this corresponds to determining the reaction type associated with a given set of reactants and products and their topology.
>
> We acknowledge that DLGNet could be applied to link prediction. However, the focus of this paper is hyperedge classification, and link prediction falls outside the scope of our study. We do not understand why the reviewer refers to performance on a task that we did not test our method on.
> If the reviewer is referring to hyperlink prediction, it is an interesting topic that we would like to explore in future work.
>
> [1] Daud, Nur Nasuha, et al. "Applications of link prediction in social networks: A review." Journal of Network and Computer Applications 166 (2020): 102716.
>
> **Lack of details of how to model the reaction structures to hypergraph edge prediction task. Is it the node of the hypergraph is the molecular? This problem definition should be clearly shown in the main paper.**
>
> Section 1, from line 57 to line 97, describes the problem modeling. Specifically, each molecule is modeled as a node within the hypergraph, and interactions between molecules are modeled as directed hyperedges. Given that we know the reactants and products in advance, we can assign a direction to the hyperedges, as illustrated in Figures 1 and 2 of the main paper. To clarify this aspect, we have changed Figure 1 (and the caption) and added a new reference to this description in Section 5.3.
>
> **Further, why DLG design can specifically work for this reaction task? In our understanding, the specific application paper should consider some domain knowledge (such as chemical reaction prior information) to make the proposed approach convincing.**
>
> Our proposed model considers domain knowledge such as chemical reaction prior information: every chemical reaction present in the dataset is topologically encoded by a hyperedge, reagent and product molecules are encoded as nodes, and Morgan Fingerprints are used to determine the node features; also,  chemical-reaction class labels as used to train the model.

---

> ### Author Response · Authors · 2024-11-20
> **Response to Reviewer 2 (parts 2/3)**
>
> **The design of the $\vec B_{ve} = -i$ if $v \in T(e)$ is weird, what is the theoretical or intuitive motivation? What if we use $\vec B_{ve} = -1$ if $v \in T(e)$ or we still use the B as described in the typical method like [1].**
>
> As explained in Section 3 and demonstrated with an example in Appendix G, utilizing an incidence matrix with complex numbers, allows us to encode directionality and construct a Laplacian, the Directed Line Graph Laplacian, that is Hermitian and meets the necessary properties for applying a spectral-based approach (see Section 3 and Appendix B for further details).
>
> If, instead, we had chosen to use $B_{ve} = -1$, we would have lost the directionality of the hypergraph. To illustrate, consider (for simplicity---this can be observed also for more hypergraphs) a graph with nodes $1,2$ and edges $e_1 = (2,1)$ and $e_2 = (3,2)$.
> We have $\vec{B} = \begin{pmatrix} -i & 0 \\\\ 1 & -i \\\\ 0 & 1 \end{pmatrix}$ and $B = \begin{pmatrix} -1 & 0 \\\\ 1 & -1 \\\\ 0 & 1 \end{pmatrix}$. The Laplacian matrix we use in DLGNet is  $\vec{L} = \begin{pmatrix} 2 & -i \\\\ i & 2 \end{pmatrix}$. The Laplacian matrix using $B$ reads $L = \begin{pmatrix} 2 & -1 \\\\ 1 & 2 \end{pmatrix}$. Let us recall that the nodes of both matrices correspond to the edges of the graph. Therefore, $\vec L_{12}$ indicates the presence of the directed line graph edge $(e_1, e_2)$, which captures the topology of the original graph where edge $e_2$ is seen before edge $e_1$ in a path from node 3 to node 1. Differently, since $L_{12} = L_{21}$, in $L$ such directional information is completely lost.
>
> Regarding the solution proposed in [1], this approach defines two separate incidence matrices: one for tail elements $B_T=\begin{pmatrix}1&0\\\\0&1\\\\0&0\end{pmatrix}$ and one for head elements $B_H=\begin{pmatrix}0&0\\\\1&0\\\\0&1\end{pmatrix}$. The Laplacian matrix using $B_T$ and $B_H$ reads
> $L=\begin{pmatrix}0&0\\\\1&0\end{pmatrix}$. This matrix $L$ is not symmetric, which does not fit the framework in which we operate in our paper, in which we are designing a spectral-based convolutional operator. This type of GNN does not permit the use of a non-symmetric matrix, as it requires an eigenvalue decomposition of the Laplacian matrix with real eigenvalues. We note that, thanks to the adoption of a complex-valued $\vec B$, our proposed Laplacian matrix is Hermitian and, therefore, admits the required eigenvalue decomposition.
> We have added this explanation to the appendix of the paper, in section H.
>
> [1] Directed Hypergraph Representation Learning for Link Prediction, AISTATS 2024

---

> ### Author Response · Authors · 2024-11-20
> **Response to Reviewer 2 (parts 3/3)**
>
> **This paper lacks some necessary baselines and is without discussion with some related works. For example [1,2,3]. Besides, the Magnetic Laplacian also be used in [4].**
>
> In response to the feedback, we have expanded the discussion of related work in the main paper (see Section 5.2).
>
> We also explored the possibility of incorporating the models suggested by the reviewer, in addition to the already substantial set of 12 state-of-the-art models used as baselines. Below are some reflections that resulted from this activity.
>
> 1. As for the model used in [1], the architecture proposed is designed to solve the link prediction problem, whereas, in this paper, we are facing the hyperedge classification task. However, we were able to adapt the algorithm to our specific task and we obtained the following results:
>
> **Table:** Mean F1-score and standard deviation obtained on the hyperedge classification task.
>
> | Topology            | Method     | Dataset-1        | Dataset-2        | Dataset-3        |
> | ------------------- | ---------- | ---------------- | ---------------- | ---------------- |
> | Directed Hypergraph | DHRL [1]   | 58.15 ± 1.58     | 79.36 ± 3.94     | 99.27 ± 0.79     |
> | Directed-Line Graph | **DLGNet** | **60.55 ± 0.80** | **83.67 ± 3.41** | **99.75 ± 0.34** |
>
> As shown in the table, our method (DLGNet) outperforms DHRL.
>
>
> 2. The method in [2], proposed in 2022, is an undirected HNN. Since there are already nine state-of-the-art models in the undirected HGNN field—including two more recent approaches than [2], ED-HNN and PhenomNN, which have become a standard benchmark for HNN—we believe an additional comparison is unnecessary. Furthermore, we believe that the recent inclusion of [1] (mentioned by the reviewer) and our pre-existing comparison with DHM sufficiently address the relevant advancements in this field.
>
> 3. The paper in [3] is a preprint available on arXiv, meaning it has not undergone formal peer review. Additionally, no GitHub code has been provided. Moreover, numerous methods have been developed after its publication on arXiv, such as HCHA, HCHA with the attention mechanism, HNHN, HyperGCN, UniGCNII, HyperDN, AllDeepSets, AllSetTransformer, LEGCN, ED-HNN, PhenomNN, and DHM, which do not include [3] in their comparisons.
>
> 4. As for the model used in [4], since in our work we propose a different Laplacian, the Directed Line Graph Laplacian, which is distinct from the Magnetic Laplacian. Moreover, in [4], a Random Walk-based Hypergraph Laplacian is used, not the Magnetic Laplacian as pointed out by the reviewer.
>
> [1] Directed Hypergraph Representation Learning for Link Prediction, AISTATS 2024.\
> [2] Hypergraph Convolutional Networks via Equivalency between Hypergraphs and Undirected Graphs, ICML workshop 2022.\
> [3] Directed hypergraph neural network, 2020.\
> [4] Unified Random Walk, Its Induced Laplacians and Spectral Convolutions for Deep Hypergraph Learning, 2022

---

> > ### Comment · Reviewer_nufw · 2024-11-29
> > **Some related work should be dissused**
> >
> > Thanks for the response, I am still concerned about the following question.
> > 1. The proposed method is designed for directed hypergraph edge prediction/ classification, so it is vital to know the disadvantages of the existing directed hypergraph methods like [3], although you don't want to implement them. Using the undirected hypergraph NNs to be the baselines sounds unsupported, as we can see the poor performance, so more stronger baseline should be considered.
> >
> > 2. The authors claim that no traditional methods exist to process the reaction classification problem. I'm not an expert in chemistry, but I find some related work like[5]. Further, I also asked for the chemists and they said template matching may be the traditional method. If there did not exist traditional works, I wonder about the real meaning of this task. How to use your method in the real chemistry industry.  Further, how about considering using the directed graph neural network to process it?
> >
> > 3. figure 1 should tell the reader what a, b.c represent.
> > [3] Directed hypergraph neural network, 2020.
> > [5] Structure-Based Classification of Chemical Reactions without Assignment of Reaction Centers, JCIN 2005

---

> ### Comment · Area_Chair_B1Yo · 2024-11-28
> **Please acknowledge authors' rebuttal**
>
> Hi Reviewer nufw,
>
> The authors made great efforts to address your concern, but they have not heard back from you yet. Now the discussion stage is ending soon, and I hope you can take some time to acknowledge the authors' response.
>
> At your earliest convenience, could you check the response from the authors, and then indicate if it changes your opinions?
>
> Best,
>
> AC

---

> ### Author Response · Authors · 2024-11-29
> **Response to Reviewer 2 (parts 1/2)**
>
> **The proposed method is designed for directed hypergraph edge prediction/ classification, so it is vital to know the disadvantages of the existing directed hypergraph methods like [3], although you don't want to implement them.**
>
> To demonstrate the effectiveness of our method on the hyperedge classification task, we compare it against 12 baselines, including two that utilize directed hypergraphs. Although [3] is an ArXiv preprint without peer review or publicly available code, we reimplemented the model, and the corresponding results are reported in the table below.
>
> **Table:** Mean F1-score and standard deviation obtained on the hyperedge classification task.
>
> | Topology            | Method     | Dataset-1        | Dataset-2        | Dataset-3        |
> | ------------------- | ---------- | ---------------- | ---------------- | ---------------- |
> | Directed Hypergraph | DHNN [3]   | 58.19 ± 0.82     | 79.75 ± 2.46     | 99.51 ± 0.51     |
> | Directed-Line Graph | **DLGNet** | **60.55 ± 0.80** | **83.67 ± 3.41** | **99.75 ± 0.34** |
>
> In this comparison, our method clearly outperforms DHNN [3] for the given task.
>
> [1, 3] feature the same Laplacian matrix. These works define a Laplacian matrix that relies on the personalized PageRank matrix $P$ to encode directionality. This approach is inspired by [8], which applied a similar technique to directed graphs. In contrast, our proposed Laplacian, the Directed Line Graph Laplacian, is defined for directed hypergraphs via a transformation to directed line graphs with complex weights, using explicitly complex numbers to encode directionality. Thus, the difference lies not only in how directionality is encoded but also in the structures to which the two Laplacians are applied: our Laplacian is designed for the directed line graph, while theirs is tailored to the directed hypergraph.
> The computational experiments that we added show that our approach leads to better performance for the task we considered.
>
> **Using the undirected hypergraph NNs to be the baselines sounds unsupported, as we can see the poor performance, so more stronger baseline should be considered.**
>
> We would like to clarify that the comparison with undirected hypergraphs is included to demonstrate the importance of directionality in this task.
>
> Furthermore, we didn't compare our proposed only to undirected methods. Indeed, we compared our approach against three methods, including [3], which utilize directed hypergraphs.
> Finally, to ensure the comprehensiveness and rigor of the experimental setup, we conducted an ablation study and performed a qualitative analysis of the results.
>
> **The authors claim that no traditional methods exist to process the reaction classification problem. I'm not an expert in chemistry, but I find some related work like[5]. Further, I also asked for the chemists and they said template matching may be the traditional method.**
>
> Thank you very much for bringing [5] to our attention. This 2005 paper uses classical machine learning techniques (Random Forests), which is reasonable since deep learning techniques were in their infancy in 2005. The study builds on MOLecular Maps of Atom-level Properties (MOLMAPs) as descriptors, which, while insightful, depend on the accurate 3D structures of molecules. Obtaining such structures requires computationally intensive processes, including energy minimization via Force Fields or quantum-mechanical calculations. Consequently, MOLMAPs see limited use in chemistry applications like molecular property prediction, where 2D topological descriptors, which are computationally simpler, are typically preferred.
>
> Template matching is not related to our application; instead, it refers to a retrieval method aiming at finding molecules containing certain structural features, such as pharmacophores or functional groups (for example a benzene ring, by explicitly using a benzene ring as the template).
>
>
> [1] Directed Hypergraph Representation Learning for Link Prediction, AISTATS 2024.
>
> [3] Directed hypergraph neural network, 2020.
>
> [5] Structure-Based Classification of Chemical Reactions without Assignment of Reaction Centers, JCIN 2005
>
> [6] Chemical space: limits, evolution and modelling of an object bigger than our
> universal library. Digital Discovery, 1(5):568–585, 2022
>
> [7] Dhmconv: Directed hypergraph momentum convolution framework. In International Conference on Artificial Intelligence and Statistics, PMLR, 2024
>
> [8] Digraph inception convolutional networks, Advances in neural information processing systems 33, 2020.

---

> ### Author Response · Authors · 2024-11-29
> **Response to Reviewer 2 (parts 2/2)**
>
> **If there did not exist traditional works, I wonder about the real meaning of this task. How to use your method in the real chemistry industry.**
>
> Regarding the significance of the task, classifying chemical reactions into well-defined classes, including nucleophilic substitution, addition, etc., is fundamental for a systematic understanding of chemical reactivity, enabling researchers to predict outcomes across related systems and support reaction design by drawing analogies with known transformations.
>
> The main practical applications in industrial chemistry of the task that we tackle in our paper, i.e., accurately classifying chemical reactions, are found in key areas such as drug discovery and retrosynthesis planning, where understanding reaction classes involved in synthesizing complex molecules can aid in selecting the appropriate synthetic routes, especially for steps requiring stereoselective or regioselective transformations.
> Solving such a chemical reaction classification task can help narrow down the range of possible transformations to only those applicable to a given class, thereby improving both speed and reliability.
>
> In a synthetic strategy, accurately classifying reactions can suggest the use of specific catalysts or solvents to optimize yields. In fact, tailoring the catalyst to the reaction class of interest is central in the field of catalyst development, and plays a crucial role in improving efficiency in areas such as polymerization or cross-coupling.
>
> Reaction class classification is not just a theoretical endeavor, but it is meant to provide a structured framework with direct benefits for research, development and production.
>
> **Further, how about considering using the directed graph neural network to process it?**
>
> We thank the reviewer for their question. We want to highlight that DLGNet is a directed Graph Neural Network.
>
> Nevertheless, if the reviewer is referring to the use of another directed GNN on the directed line graph, we would like to emphasize that the transformation from a directed hypergraph to a directed line graph was introduced in this paper. By definition (see Definition 1 in the main paper), the adjacency matrix of the directed line graph (Equation (8)) contains complex numbers, as it encodes the directionality present in the directed hypergraph. This results in a Hermitian adjacency matrix, a property that, to the best of our knowledge, has not yet been addressed by existing GNN models, which lack the ability to define a Laplacian starting from such an adjacency matrix.
>
> If, on the other hand, the reviewer is referring to the use of directed GNNs to model the interaction between reactants and products as a directed graph, as explicitly stated in lines 59–61, modeling a reaction as a set of individually directed edges fails to capture the intricate complexity of these interactions [6]. Representing the reaction as an undirected hypergraph provides an initial means to account for the entirety of the relationships involved but falls short in encoding the directionality of the reaction. Therefore, we propose the use of a directed hypergraph as the most suitable approach to capture both the complexity of the reactant/product interactions and their inherent directionality.
>
>
> **figure 1 should tell the reader what a, b.c represent.**
>
> The symbols $a, b, c, d$, and $e$ represent five distinct molecules, while $e_1, e_2$, and $e_3$ are three reactions. We will include this clarification in the caption of Figure 1.
>
> [1] Directed Hypergraph Representation Learning for Link Prediction, AISTATS 2024.
>
> [3] Directed hypergraph neural network, 2020.
>
> [5] Structure-Based Classification of Chemical Reactions without Assignment of Reaction Centers, JCIN 2005
>
> [6] Chemical space: limits, evolution and modelling of an object bigger than our
> universal library. Digital Discovery, 1(5):568–585, 2022
>
> [7] Dhmconv: Directed hypergraph momentum convolution framework. In International Conference on Artificial Intelligence and Statistics, PMLR, 2024
>
> [8] Digraph inception convolutional networks, Advances in neural information processing systems 33, 2020.

---

### Official Review · Reviewer_zD8y · 2024-11-04

**Soundness:** 2
**Presentation:** 2
**Contribution:** 2
**Rating:** 5
**Confidence:** 4

**Summary:**

This paper investigates the problem of chemical reaction classification based on hypergraphs. Firstly, this paper introduces a formal definition of directed line graphs to transform directed hypergraphs, thereby converting edge-level tasks into node-level tasks. Building on this definition, DLGNet is proposed, which updates hyperedge features through local aggregation. Experimental results demonstrate the superior performance of the proposed DLGNet compared to the baseline chemical reaction classification.

**Strengths:**

1) The picture of modeling chemical reactions using hypergraphs is interesting in GNNs for chemistry and biology.

2)  Good performance compared to the "baseline".

**Weaknesses:**

1) The novel of this paper is limited. On the one hand, the transformation of directed graphs to line graphs has been extensively studied, and the proposed DLG does not capture the unique characteristics that distinguish hypergraphs from ordinary graphs. Additionally, the use of complex numbers to represent directionality in directed graph neural networks (such as MagNet [1]) has already been explored. Thus, the contribution of this part appears to be incremental. On the other hand, it is unclear what the rationale for translating directed hypergraphs into directed line graphs for hyperedge classification is.

2) The paper is poorly organized. Is the introduction of Datasets 1, 2, and 3 a contribution to this paper? Why devote so much space to this in the main text?

3) The discussion and comparison of related work are insufficient. In particular, all compared models are not specific to chemical reaction classification.

[1] MagNet: A Neural Network for Directed Graphs. NeurIPS 2021

**Questions:**

Refer to Weaknesses.

---

> ### Author Response · Authors · 2024-11-20
> **Response to Reviewer 1 (parts 1/2)**
>
> We thank the reviewer for reviewing our paper and providing their comments.
>
> **The novel of this paper is limited. On the one hand, the transformation of directed graphs to line graphs has been extensively studied, and the proposed DLG does not capture the unique characteristics that distinguish hypergraphs from ordinary graphs. Additionally, the use of complex numbers to represent directionality in directed graph neural networks (such as MagNet [1]) has already been explored. Thus, the contribution of this part appears to be incremental. On the other hand, it is unclear what the rationale for translating directed hypergraphs into directed line graphs for hyperedge classification is.**
>
> We disagree that our model lacks novelty simply because it incorporates complex numbers to encode directionality. Our Laplacian, the Directed Line Graph Laplacian, has not been proposed in prior work. While we do employ complex numbers as MagNet [1] (and other proposals [3 , 4, 5] published in prestigious venues), this is the only thing the two approaches have in common.
> As a matter of fact, our proposal is grounded on a change of perspective through the definition of a directed line graph generated from an incidence matrix specifically defined for directed hypergraphs. To the best of our knowledge, this is the first formal definition of such an operation.
> The use of such a graph (2-uniform hypergraph) enables the use of spectral graph theory, but the resulting Laplacian does not resemble the one proposed by and used in MagNet [1] (which actually originated in physics [2] and was then adapted to the GCN framework).
>
> We also fear there is a misunderstanding. As highlighted in the first contribution of our paper, our work defines a new concept by transforming a directed hypergraph into its corresponding directed line graph. Thus, we are not applying any transformation of directed graphs to line graphs, to the best of our knowledge, this is the first formal definition of such an operation.
>
> [1] MagNet: A Neural Network for Directed Graphs. NeurIPS 2021\
> [2] Fluxes, Laplacians, and Kasteleyn’s theorem. In Statistical Mechanics 1993.\
> [3] Msgnn: A spectral graph neural network based on a novel magnetic signed laplacian. LoG 2022.\
> [4] Sigmanet: One laplacian to rule them all. AAAI 2023.\
> [5] A spectral graph convolution for signed directed graphs via magnetic laplacian. Neural Networks 2023
>
> **On the other hand, it is unclear what the rationale for translating directed hypergraphs into directed line graphs for hyperedge classification is.**
>
> The hyperedge classification task could be approached by pairing the feature vectors of the nodes and passing them to a classifier. However, in the case of hypergraphs, this operation is not well-defined because the number of nodes in a hyperedge is not constant and, in principle, could require concatenation of up to $n$ node features, where $n$ is the number of nodes.
> For this reason, the hyperedge classification task can be reframed as a node classification task by using the concept of the line graph. In this formulation, the hyperedges of the original hypergraph are treated as nodes in the line graph.
> From a mathematical perspective, in an HNN (and equivalently in a GCN), the output is given by:
>
> $$
> \hat{X}' \in \mathbb{R}^{n \times c'}
> $$
>
> where $c'$ represents the number of classes to be predicted. While converting the directed hypergraph into the directed line graph, the hyperedges become the node of the line graph. Consequently, the feature output becomes:
>
> $$
>  \hat{X}'' \in \mathbb{R}^{m \times c''}
> $$
> where $m$ is the number of hyperedges in the hypergraph, and $c''$ is the number of node classes to be predicted.
>
> **The paper is poorly organized.**
>
> While we are willing to review and improve the structure of the paper, 18Zj and YKY8 were satisfied about the flow of the paper stating that is clear, rich in details, and the method is mathematically grounded. Nonetheless, we would appreciate more detailed guidelines on how improve the organization of the paper.

---

> ### Author Response · Authors · 2024-11-20
> **Response to Reviewer 1 (parts 2/2)**
>
> **Is the introduction of Datasets 1, 2, and 3 a contribution to this paper? Why devote so much space to this in the main text?**
>
> We constructed Datasets 2 and 3 by adapting existing datasets from the literature, so in a way, they represent a (minor) contribution to our work. For this reason, and given the focus of our paper on the chemical-reaction classification task, we consider it essential to provide a detailed description of these datasets. This not only ensures reproducibility but also highlights their potential utility for the broader chemistry community. Furthermore, a comprehensive explanation of these datasets helps clarify the nature of the task, making it more accessible and relevant for researchers in the AI community. However, due to the reviewer's concern, we have reduced the length of this part, moving some details to Appendix D.
>
> **The discussion and comparison of related work are insufficient. In particular, all compared models are not specific to chemical reaction classification.**
>
> We evaluated DLGNet against 12 baselines, which we believe constitutes a robust comparison. In response to the feedback, we have expanded the discussion of related work in the main paper (see Section 5.2). However, we were not aware of any specific models for the task. If the reviewer knows of any, we would be open to discussing them.

---

> > ### Comment · Reviewer_zD8y · 2024-11-25
> >
> > Thanks to the authors for their response. I have carefully read your response and the revised manuscript. This solved some of my concerns. Thus, I would like to raise the rating from 3 to 5. Considering the limited novelty, this rating will not improve further.

---

> > > ### Author Response · Authors · 2024-11-25
> > >
> > > We thank the reviewer for reconsidering their initial score. However, we would like to stress that we do not agree on the novelty of our paper being limited. Our Directed Line Graph Laplacian is a novel Laplacian, uniquely capable of operating directly on directed and undirected line graphs, and enjoying many relevant mathematical properties which we established with careful proofs. There is a large stream of papers in the GCN literature published in the proceedings of top AI conferences that propose novel Laplacian and/or convolution operators by generalizing previous proposals or extending them to more general cases, and ours is well aligned with them.

---

### Meta-Review · Area_Chair_B1Yo · 2024-12-19

**Metareview:**

This paper proposes a new GNN-based method DMLNet for chemical reaction classification, by transforming a directed hypergraph into the corresponding directed line graph. A spectral-based GNN is developed that uses a Hermitian Laplacian matrix to encode the directionality and connectivity between hyperedges, which is suitable for spectral convolution, and thus enhances classification performance. Currently, the paper receives a mix of opinions from reviewers. The main concern lies in several aspects, including poor presentation, limited novelty, the inputs being just molecular Morgan fingerprints without even atom type or other information, etc. Considering these, I suggest the authors significantly improve the presentation of the paper, clearly state the difference with existing works like MagNet, and consider necessary molecular information in problem formulation and experiments (at least atom types).

**Additional Comments On Reviewer Discussion:**

The authors seemed to make great efforts to address the concerns of the reviewers. I am not confident enough that all changes have been reflected in the manuscript because the changes are not marked and the authors didn't mention the specific locations of all changes. I've checked by myself and feel some have been integrated but not all. Reviewer zD8y questioned the novelty and was not fully convinced by the authors. Reviewer nufw has strong concerns about the presentation.

---

### Decision · Program_Chairs · 2025-01-22

Reject